# Ultra-high performance wearable thermoelectric coolers with less materials

Ravi Anant Kishore[1,2], Amin Nozariasbmarz [3], Bed Poudel[3], Mohan Sanghadasa[4] & Shashank Priya [1,3]

Thermoelectric coolers are attracting significant attention for replacing age-old cooling and refrigeration devices. Localized cooling by wearable thermoelectric coolers will decrease the usage of traditional systems, thereby reducing global warming and providing savings on energy costs. Since human skin as well as ambient air is a poor conductor of heat, wearable thermoelectric coolers operate under huge thermally resistive environment. The external thermal resistances greatly influence thermoelectric material behavior, device design, and device performance, which presents a fundamental challenge in achieving high efficiency for on-body applications. Here, we examine the combined effect of heat source/sink thermal resistances and thermoelectric material properties on thermoelectric cooler performance. Efficient thermoelectric coolers demonstrated here can cool the human skin up to 8.2 °C below the ambient temperature (170% higher cooling than commercial modules). Cost-benefit analysis shows that cooling over material volume for our optimized thermoelectric cooler is 500% higher than that of the commercial modules.

[1] Center for Energy Harvesting Materials and Systems, Virginia Tech, Blacksburg, VA 24061, USA. [2] National Renewable Energy Laboratory, 15013 Denver West Pkwy, Golden, CO 80401, USA. [3] Department of Materials Science and Engineering, Pennsylvania State University, University Park, Pennsylvania, PA 16802, USA. [4] Aviation and Missile Center, U.S. Army Combat Capabilities Development Command, Redstone Arsenal, Alabama, AL 35898, USA. Correspondence and requests for materials should be addressed to R.A.K. (email: ravi86@vt.edu) or to S.P. (email: sup103@psu.edu)

Cooling devices currently account for nearly 20% of the total electricity consumed in buildings around the world[1]. Considering the rising demand for space cooling in the residential sector, global energy use for space cooling is projected to increase from 2020 terawatt hours (TWh) in 2016 to 6200 TWh in 2050[1]. The huge electricity consumption for space cooling results in substantial $CO_2$ emissions associated with the fossil fuels used in power generation. Space cooling by means of air conditioners (ACs) is also reported to be a major contributor to the urban heat island[2]. In addition, ACs use refrigerants that usually contain hydrofluorocarbons (HFCs), which contribute to global warming if leaked in the atmosphere[3]. Among all the alternatives for replacing traditional cooling technologies, thermoelectric (TE) cooling is the most promising option as it provides pathway for personalized cooling. A TE cooler (TEC) is a solid-state heat pump based on Peltier effect that transfers heat from one side of the device to the other[4,5]. In the past few decades, TECs have gained huge attention in the consumer market due to their reliable, lightweight, and noiseless operation[6]. In addition, TECs have no moving parts, utilize no working fluid, involve no chemical reactions, and produce no emissions[7,8]. These features provide TECs several competitive advantages over traditional refrigerators and other cooling technologies. TECs are, therefore, widely used for heat management in electronic packages[9,10], microprocessors[11], precision devices[12], and laser equipment[13]. Recently, several commercial products for personalized cooling, such as Embr wave TE wristband[14], Wristify TE bracelet[15], Flowtherm TE vest[16], ClimaWare TE jacket[16], and USB Forehead Neck cooler[17], have been launched. TECs are also used in modern car seat cushions[18], mattresses[19], and thermal pads[20], which experience bodily contact.

Human skin is made up of three layers: epidermis, dermis, and hypodermis[21]. Hypodermis, the deepest layer, is the subcutaneous fat layer, which is thermally insulating and is naturally designed to keep the body warm[21]. Human skin, therefore, is a poor conductor of heat ($\kappa \approx 0.3$ W/m K)[22]. In addition, in most cases, the wearable TECs need to operate in the ambient air, which has very low thermal conductivity ($\kappa \approx 0.032$ W/m K at 27 °C)[23]. The wearable TECs, therefore, experience a very low heat load and an extremely high thermally resistive environment. This presents a fundamental challenge in achieving high efficiency in TEC modules for on-body applications. The commercial TECs are typically designed to maximize the cooling in a high-heat load (>1 W/cm$^2$) conditions and for low-thermally resistive environment[24,25]. This implies that the currently available commercial TECs might not be the best choice for the wearable cooling devices. Prior studies have reported a very strong influence of external thermal resistance on the performance of TECs[26–28], and suggested that the optimal TEC should have internal thermal resistance in the range of 40–70% of the total thermal resistance of the system[29]. However, most of these studies were focused on electronic cooling using TECs. For on-body applications, the research in the field of TE, so far, has been focused on thermal energy harvesting from body heat[30–36]. The TE modules with fill factor (FF) less than 20% have been recommended for on-body applications, which brings challenges in terms of mechanical stability[36]. TE modules for on-body cooling applications have not been much explored in the literature. A flexible TE system recently reported for human body demonstrated a temperature drop of 4 °C, indicating the feasibility of using TECs to control the temperature of the human body[37]. While there are several commercial TEC products in the market, their material composition and design are proprietary.

In this paper, we provide fundamental insight on designing TECs that can be deployed in different thermally resistive environments including the on-body conditions. The combined effect of each TE material property and the resistive condition on TEC performance is described in conjunction with TEC design parameters (leg dimensions, FF, aspect ratio (AR), etc.). Using numerical and experimental studies, different TEC design architectures are investigated, specifically for on-body applications, under different ambient conditions and an optimal design for wearable TECs is obtained. The ultra-low fill factor module proposed in this paper exhibits 170% higher temperature cooling than the commercial TEC, while utilizing 500% less TE materials.

## Results

**Personalized cooling using wearable TE modules.** Wearable TEC imparts users the ability to adjust the cooling by reducing the skin temperature below ambient temperature, a sensation similar to holding an ice cube. As shown in Fig. 1a, a wearable TEC is usually accompanied with a heat sink and is placed on the skin at different parts of human body, preferably on an area exposed to the ambient atmosphere. Human hand is typically a preferred location as hand movement increases the heat transfer between heat sink and ambient, which finally results in better performance and thus enhanced cooling by the TEC. TECs require significantly less electrical power than space cooling and this technology is projected to provide huge energy cost savings per year[15]. A TEC module consists of several thermocouples, comprising of p- and n-type TE materials, connected electrically in series and thermally in parallel (Fig. 1b). Upon applying the electric current, charge carriers (electrons and holes) carry the heat from one side of the module to other and the thermocouple junctions are heated or cooled depending upon the relative direction between the current and the p–n junctions, thereby allowing heat exchange with the TEC module and the surroundings[38] (Fig. 1c). Thermodynamically, the human body is a low-temperature heat reservoir ($T_{body} \approx 37$ °C) that emits thermal energy at the rate of ~25 mW/cm$^2$ into the ambient atmosphere[39,40]. As shown in Fig. 1d, the human skin temperature, however, varies at different parts of body. The skin temperature typically increases with increase in ambient temperature and reduces from head to feet. For instance, under the normal ambient temperature condition of 23 °C, the human skin temperature is noted to be ~34 °C at the head and ~25 °C at the feet. This increases to ~36 °C at the head and ~35 °C at the feet as the ambient temperature increases to 34 °C[41]. Varying operating conditions and extremely high thermally resistive environment present difficulties in achieving high-efficiency TEC modules for on-body applications. In the following sections, various aspects of TE cooling including the combined effect of material parameters and device configuration on TEC performance are extensively studied.

**Effect of thermally resistive environment on TE materials.** There are three key TE material properties: Seebeck coefficient ($\alpha$), electrical conductivity ($\sigma$), and thermal conductivity ($\kappa$) that constitute a dimensionless metric called TE figure-of-merit, $zT$, which is defined as[42,43]:

$$zT = \frac{\alpha^2 \sigma}{\kappa} T. \tag{1}$$

It is perceived that increasing $zT$ improves the performance of TE devices; therefore, tremendous research efforts have been made in the past few decades to enhance $zT$ of TE materials[44–47]. Since $zT$ depends on three different material properties, it can be theoretically improved in multiple ways, such as increasing Seebeck coefficient, increasing electrical conductivity, lowering thermal conductivity, or combination of these three approaches. In practice, however, the three transport parameters ($\alpha$, $\sigma$, and $\kappa$)

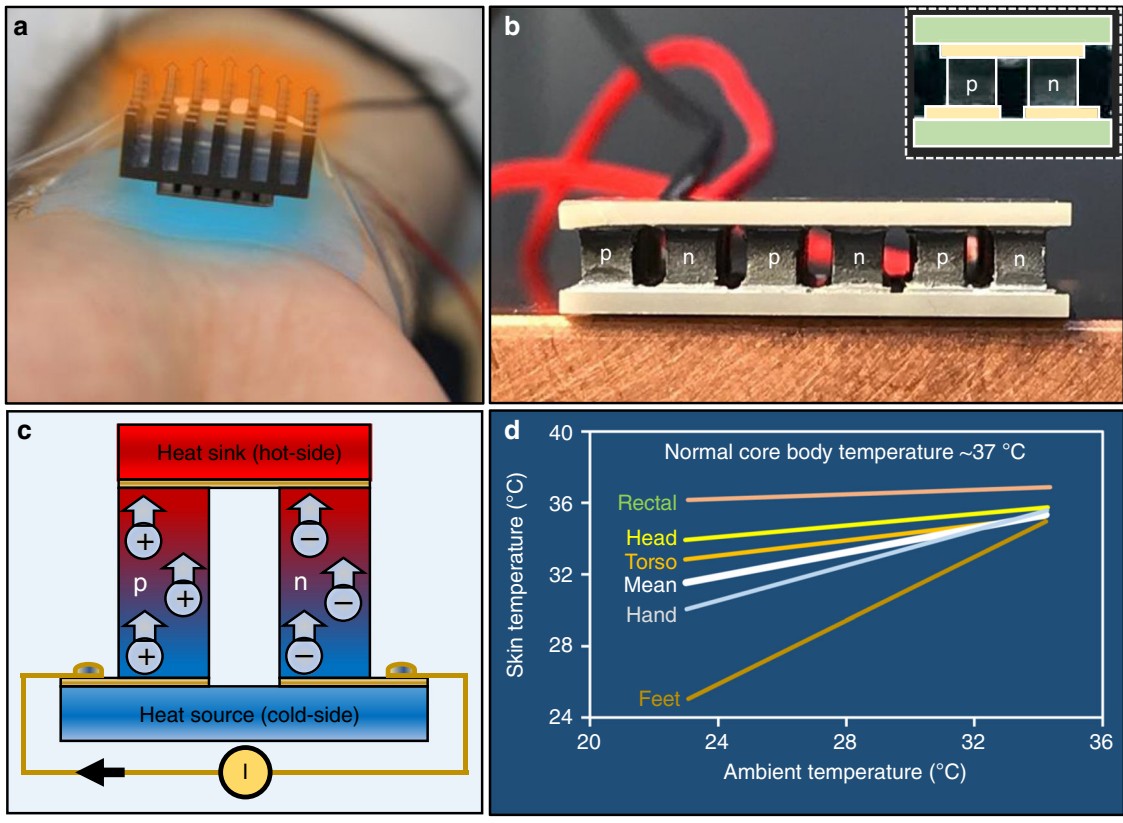

**Fig. 1** Personalized cooling using wearable TE modules. **a** Wearable thermoelectric cooler (TEC) module provides localized body cooling, thereby making space cooling unnecessary. Arrows illustrate heat flow from human body to the ambient via TEC and heat sink. **b** TECs consist of several thermocouples, comprising of p- and n-type thermoelectric legs, connected electrically in series and thermally in parallel. **c** Upon applying the electric current, charge carriers, holes in p-type and electrons in n-type legs, carry heat from one side of the module to the other, thereby allowing heat exchange with the TEC module and the surroundings. Arrows illustrate movement of charge carriers, whereas circles with positive and negative signs depict holes and electrons, respectively. **d** The human skin temperature varies at different parts of body and increases with increase in ambient temperature. Information taken from ref. [41]. By natural evolution, human skin is poor conductor of heat and so is the ambient air. The wearable TECs, therefore, operate in an extremely high thermally resistive environment

are strongly interrelated[48], which limits $zT$ value close to unity in a bulk material[49]. Using recent advances in nanotechnology, the transport parameters can be somewhat decoupled, providing room temperature $zT$ up to 2.4 in quantum dot superlattice TE materials[50,51]. Despite these discoveries, the performance of TE devices during real deployments has remained poor, indicating a strong influence of operating environment on the device performance[29,52]. Traditionally, the performance of a TEC is measured in terms of cooling capacity which is defined as the maximum heat transfer rate from cold-side to hot-side at certain temperature difference. Theoretically, it is expected that increasing Seebeck coefficient and electrical conductivity and lowering thermal conductivity of TE materials should enhance the cooling performance of TECs. In order to examine the influence of thermally resistive environment on TE material behavior, we have considered four systematic models of TEC modules, which are listed in Supplementary Table 1. All four TEC modules are dimensionally identical; however, TEC 1 is built of TE materials with $zT = 0.62$ and other three TECs have TE materials with $zT = 1.25$ at 25 °C. Higher $zT$ of the TE materials in TEC 2, TEC 3, and TEC 4 is achieved by increasing Seebeck coefficient, increasing electrical conductivity, and reducing thermal conductivity, respectively. Figure 2 compares the cooling capacity of these TECs operating under various thermally resistive environments. In all the considered cases, heat source is maintained at 22 °C, whereas heat sink is maintained at 27 °C. Figure 2a depicts

an ideal case when heat source thermal resistance ($R_{source}$) and heat sink resistance ($R_{sink}$) are zero, which leads to $T_c = T_{source}$ and $T_h = T_{sink}$. Figure 2b–d illustrates the more realistic conditions when $R_{source} = R_{sink} = \frac{1}{hA} \neq 0$, where $A$ is the base area of TEC module in contact with the heat source and sink and $h$ is the heat transfer coefficient. Depending on the kind of medium used in the heat exchangers, the value of $h$ can differ, as shown in Supplementary Table 2.

It can be noted from Fig. 2a that, when there is no heat source or sink resistance ($h = \infty$), increase in $zT$ by increasing Seebeck coefficient produces the maximum cooling capacity, followed by increase in electrical conductivity. Interestingly, lowering thermal conductivity has no major impact on the cooling capacity. This is also true when $h = 1000$ W m$^{-2}$ K$^{-1}$ (Fig. 2b). However, it is important to notice that the effect of TE material thermal conductivity becomes increasingly important as heat source and sink resistances increase. When $h = 500$ W m$^{-2}$ K$^{-1}$, the maximum cooling capacities through high-Seebeck coefficient and low-thermal conductivity are almost the same (Fig. 2c). As source and sink resistances become extremely large ($h = 100$ W m$^{-2}$ K$^{-1}$), increase in $zT$ by lowering thermal conductivity generates the maximum cooling capacity (Fig. 2d). This discussion clearly establishes that the external thermal resistances have a very strong influence on the material behavior and its effect on the TEC performance. Most importantly, $zT$ of TE material is not the only decisive parameter that dictates the TEC performance under all

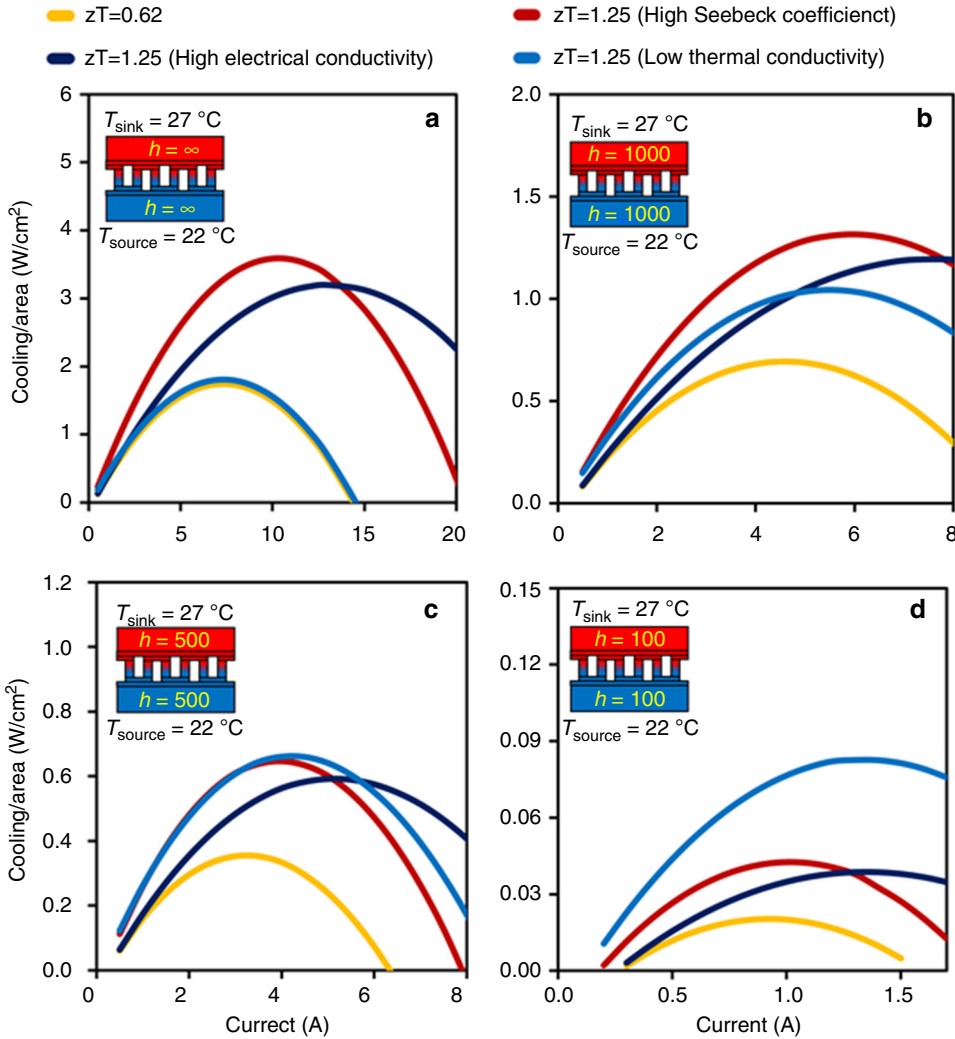

**Fig. 2** The effect of thermally resistive environment on TE material behavior. Effect of material properties on cooling capacity changes with change in the environmental condition, even if $zT$ value is the same. **a**, **b** When source/sink resistances are low ($h > 1000\,\mathrm{W\,m^{-2}\,K^{-1}}$), TE material with higher Seebeck coefficient is the best option. **c**, **d** When source and sink resistances are considerably high ($h = 100 - 500\,\mathrm{W\,m^{-2}\,K^{-1}}$), TE material with lower thermal conductivity is a better choice

operating environments. Under low-thermally resistive environments, high $zT$ along with high Seebeck coefficient is desired; whereas, under high-thermally resistive conditions, such as deployment on human body, thermal conductivity of TE materials is of utmost importance.

The effect of material properties on cold-side temperature of the TECs was also studied under different thermally resistive environmental conditions (refer Supplementary Fig. 1). As expected, when heat source/sink resistances were low ($h > 1000\,\mathrm{W\,m^{-2}\,K^{-1}}$), TE material with higher Seebeck coefficient resulted in more cooling and thus lower cold-side temperature (Supplementary Fig. 1a, b). However, when heat source and sink resistances were high ($h < 500\,\mathrm{W\,m^{-2}\,K^{-1}}$), low-thermal conductivity caused the least cold-side temperature (Supplementary Fig. 1c, d). We also studied the effect of thermally resistive environment on coefficient of performance (COP), which is illustrated in Supplementary Fig. 2. COP is defined as cooling capacity per unit input electrical power. It can be noted that for COP, electrical conductivity seems to be the most important material property when heat source and sink resistances are negligible (Supplementary Fig. 2a–c). However, under high-thermally resistive environment, low-thermal conductivity results in superior COP (Supplementary Fig. 2d).

**Effect of resistive environment on the TEC design**. The internal electrical and thermal resistances of TEC are dependent on the number and dimensions of TEC legs (see Eqs. (16) and (17) in Methods section). Therefore, variation in TEC design in conjunction with the external thermal resistances may result in huge variation in TEC performance. Traditionally, two dimensionless parameters called AR and FF are defined to account for the leg features, which are given as:

$$\mathrm{AR} = \frac{l}{w}, \qquad (2)$$

$$\mathrm{FF} = \frac{Nw^2}{A}, \qquad (3)$$

where $l$ is leg height, $w$ is the width, $N$ is total number of legs, and $A$ is the module base area. Figure 3 illustrates the effect of varying AR and FF on TEC performance under two resistive environmental conditions ($h = 1000\,\mathrm{W\,m^{-2}\,K^{-1}}$ and $h = 100\,\mathrm{W\,m^{-2}\,K^{-1}}$). The first scenario with $h = 1000\,\mathrm{W\,m^{-2}K^{-1}}$ represents the low-thermally resistive environment, while the second scenario with $h = 100\,\mathrm{W\,m^{-2}\,K^{-1}}$ signifies the high-thermally resistive environment. Heat source temperature is fixed at 22 °C and heat sink is maintained at 27 °C. The TECs considered in this section utilize

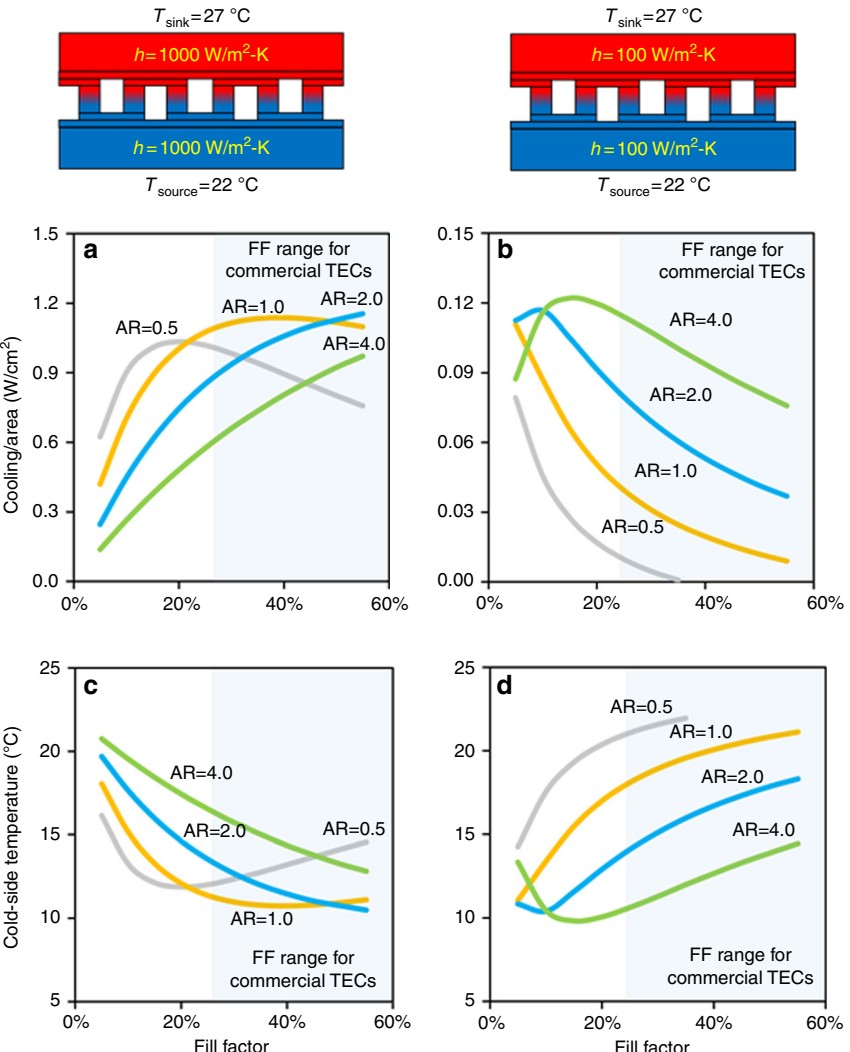

**Fig. 3** The effect of resistive environment on the optimal design of TECs. The optimal fill factor (FF) and aspect ratio (AR) are strongly affected by heat source and sink resistances. **a** When heat source and sink resistances are low ($h = 1000$ W m$^{-2}$ K$^{-1}$), the optimal FF is more than 20%. **b** When heat source and sink resistances are high ($h = 100$ W$^{-2}$ K$^{-1}$), the optimal FF is less than 20%. **c**, **d** The variation in cold-side temperature with variation in FF and AR follows the same trend as cooling capacity

commercial p- and n-type bismuth telluride materials, whose temperature-dependent material properties are provided in Supplementary Fig. 3 and Supplementary Fig. 4. It is evident from Fig. 3a, b that the effect of AR and FF on cooling capacity in the two scenarios is not the same. When heat source and sink resistances are low ($h = 1000$ W m$^{-2}$ K$^{-1}$), the optimal FF is more than 20%; whereas when heat source and sink resistances are high ($h = 100$ W m$^{-2}$ K$^{-1}$), the optimal FF is less than 20%. In both cases, however, there exists an optimal FF where cooling capacity is maximum. The optimal FF is dependent on AR and it increases when AR is increased. It can be also noted that cooling capacity at optimal FF increases with increase in AR. However, the gain in cooling capacity is small when AR > 2. Figure 3c, d illustrates the cold-side temperature of TECs. The trends are in agreement with the cooling capacity. When heat source and sink resistances are low ($h = 1000$ W m$^{-2}$ K$^{-1}$), the cold-side temperature is minimum when the optimal FF is more than 20%; whereas when heat source and sink resistances are high ($h = 100$ W m$^{-2}$ K$^{-1}$), the least cold-side temperature occurs when the optimal FF is less than 20%.

Figure 3 also illustrates FF for commercial TECs. Commercial TECs are usually manufactured with FF greater than 25% and AR close to unity. We can note from Fig. 3a, c that the commercial

TEC design looks practical when source and sink thermal resistances are negligible. Unfortunately, when source and sink thermal resistances are large, the design of commercially available TECs is not optimal. TECs built for high-thermally resistive environment should have small FF and relatively larger AR, as noted from Fig. 3b, d. The slender legs, however, may affect the structural integrity of the TEC modules; therefore, a trade-off between performance and strength is required. These observations can be employed to design an optimal TEC module for on-body applications, which is discussed in the next section.

It should be noted that traditionally cooling capacity (W m$^{-2}$) and COP are used in the literature to characterize TECs (refer Supplementary Note 2 for more discussion). For body cooling, however, cold-side temperature and temperature drop ($\Delta T$) can be considered a more appropriate criterion to judge the effectiveness of TECs as temperature change can be easily perceived by the human body. Prior studies have reported that the human skin on certain part of human hand can perceive the temperature differential as low as 0.20 °C for warming (at a rate of 2.1 °C s$^{-1}$) and 0.11 °C for cooling (at a rate of 1.9 °C s$^{-1}$)[53,54]. Therefore, in the remainder of paper, cold-side temperature and temperature drop ($\Delta T$) are reported in order to compare different

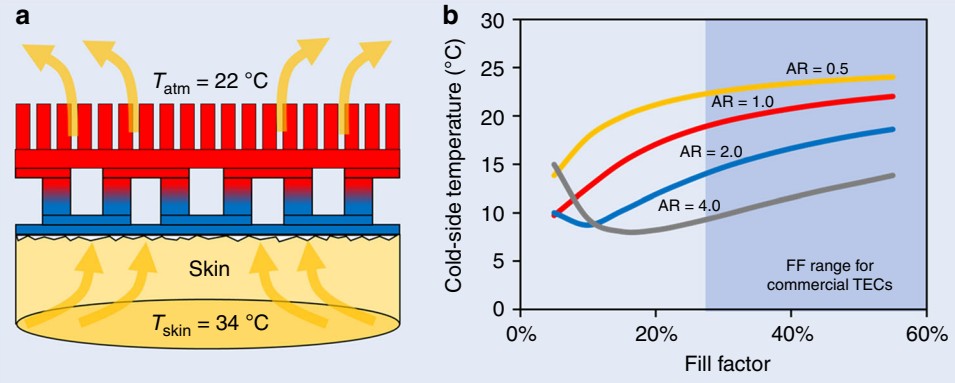

**Fig. 4** Optimal module design for wearable TECs. **a** Schematic illustrating the wearable TEC placed on human skin. High-thermal resistivity of skin and ambient air along with the contact resistance at skin–TEC interface provide a huge thermally resistive environment to the wearable TEC. Arrows illustrate heat flow from human skin to the ambient via TEC and heat sink. **b** Cold-side temperature of the wearable TECs at different FF and AR. The optimal FF for wearable TEC should be less than 15% and AR should be in the range of 1–2

TECs considered in this study. In addition, in order to evaluate the electrical power required to achieve the maximum temperature drop, a modified COP defined as temperature drop per unit input electrical power is also calculated. Lastly, considering the human comfort and economic feasibility, weight, volume, and cost of the cooling devices are equally important. Therefore, we have also reported cooling over material volume for different kinds of TECs studied in this paper.

**TEC module design for on-body applications**. Human skin, by natural evolution, is a poor conductor of heat. Depending on the physiological parameters such as weight, age, body fat, and gender, the skin resistance can greatly vary[55]. The effective heat transfer coefficient ($h_{skin}$) for the human skin under TE module has been reported to vary in the range of 20–100 W $m^{-2}$ $K^{-1}$ [56]. For a rigid TEG deployed on a human forearm, Suarez et al. used $h_{skin} = 50$ W$m^{-2}$$K^{-1}$ [36]. On the hot-side, the wearable TECs usually require fins as the heat sink to augment the heat rejection. The effective heat transfer coefficient of a heat sink ($h_{sink}$) is function of several parameters such as thermal conductivity of the fin material, base area of the heat sink, total surface area exposed for cooling, and convection coefficient of the cooling environment. Thus, $h_{sink}$ can be greatly varied by varying any of these parameters. In this study, $h_{sink} = 100$ W $m^{-2}$ $K^{-1}$ has been considered (Supplementary Note 3 provides more discussion on calculations for $h_{sink}$). In order to optimize the TEC design for wearable applications, simulations were performed at $h_{skin} = 50$ W $m^{-2}$ $K^{-1}$, $h_{sink} = 100$ W $m^{-2}$ $K^{-1}$, $T_{skin} = 34$ °C, and $T_{atm} = 22$ °C. Results are illustrated in Fig. 4. It is evident that the cold-side temperature is least when optimal FF is less than 15%. The minimum cold-side temperature decreases with increase in AR; however, the gain is not substantial when *AR* is too large (>2).

**Experimental**. In order to validate our theoretical hypothesis, three types of TEC modules were fabricated using commercial p- and n-type bismuth telluride materials. Figure 5a–c demonstrates fabricated TEC modules attached to a heat sink of size 25.4 × 25.4 × 9.5 mm³. The heat sink is made up of black-anodized aluminum and it contains 49 pins. The TEC module shown in Fig. 5a has 36 legs of dimension 1.6 mm (length) × 1.6 mm (width) × 1.6 mm (height), whereas the modules displayed in Fig. 5b, c have 12 legs of dimension 1.6 mm (length) × 1.6 mm (width) × 1.6 mm (height) and 1.05 mm (length) × 1.05 mm (width) × 1.6 mm (height), respectively. These modules were

calculated to have FF of 36%, 12%, and 5.2%, and leg AR of 1.0, 1.0, and 1.6, respectively, and are referred as high-FF TEC, low-FF TEC, and ultra-low FF TEC. Supplementary Table 3 lists the key features of the TEC modules, while detailed discussion on these modules are provided in Supplementary Note 3. Figure 5d shows a commercial module (CUI Inc., Part # CP60131H) attached to a similar heat sink. The commercial TEC module was found to have FF of 28.4% and leg AR of 1.0. Figure 5e depicts the experimental set-up for the cooling measurement. TEC module with fins is attached to a bottom aluminum plate of size 25.4 × 25.4 × 3.2 mm³, as shown in the inset. The assembly is then placed on the human forearm and the temperature of the aluminum plate is measured using a K-type thermocouple (Omega Engineering). A power supply (Keithley 2200-20-5) was used to supply electric current to the TEC module. Silicone paste was used as the thermal interface material between the TEC module and human sink, whereas silver paste was used at the interface between TEC and aluminum fins. In order to mitigate the effect of human factors on experimental results, the experiments on different TEC modules were also performed under controlled environment and the outcomes were compared with the results obtained on human body. The controlled environment consisted of a heat source at fixed temperature (34 °C) and a thermal resistor of known thermal conductivity (0.94 W $m^{-1}$ $K^{-1}$) and dimensions: 16 mm × 18 mm × 10 mm placed between heat source and TEC to mimic the thermal resistance of the human skin. Figure 5f, g depicts the transient experimental data obtained for different FF TECs under controlled environment and on human body. It can be noted that cold-side temperature and cooling (ΔT) obtained under controlled environment and on human body for different TECs follow the similar trend. Preliminary experiments revealed that the optimal electric current is ~1.0 A for the high-FF TEC, ~2.4 A for the low-FF TEC, and ~2.0 A for the ultra-low FF TEC. It can be noted that at a fixed applied current, cold-side temperature of a TEC first decreases with time, reaches a minimum value, then slowly increases and finally saturates after ~10 min. The transient minima are lower than the steady state cold-side temperature for all TECs. It is interesting to note that the ultra-low FF TEC generates the least cold-side temperature and thus maximum cooling followed by the low-FF TEC and the high-FF TEC.

Figure 6 shows the cold-side temperature and cooling (temperature drop from initial temperature) versus electric current for the different fabricated TEC modules. The experiments were repeated several times and results from four repetitions are shown in Fig. 6. The average initial temperature

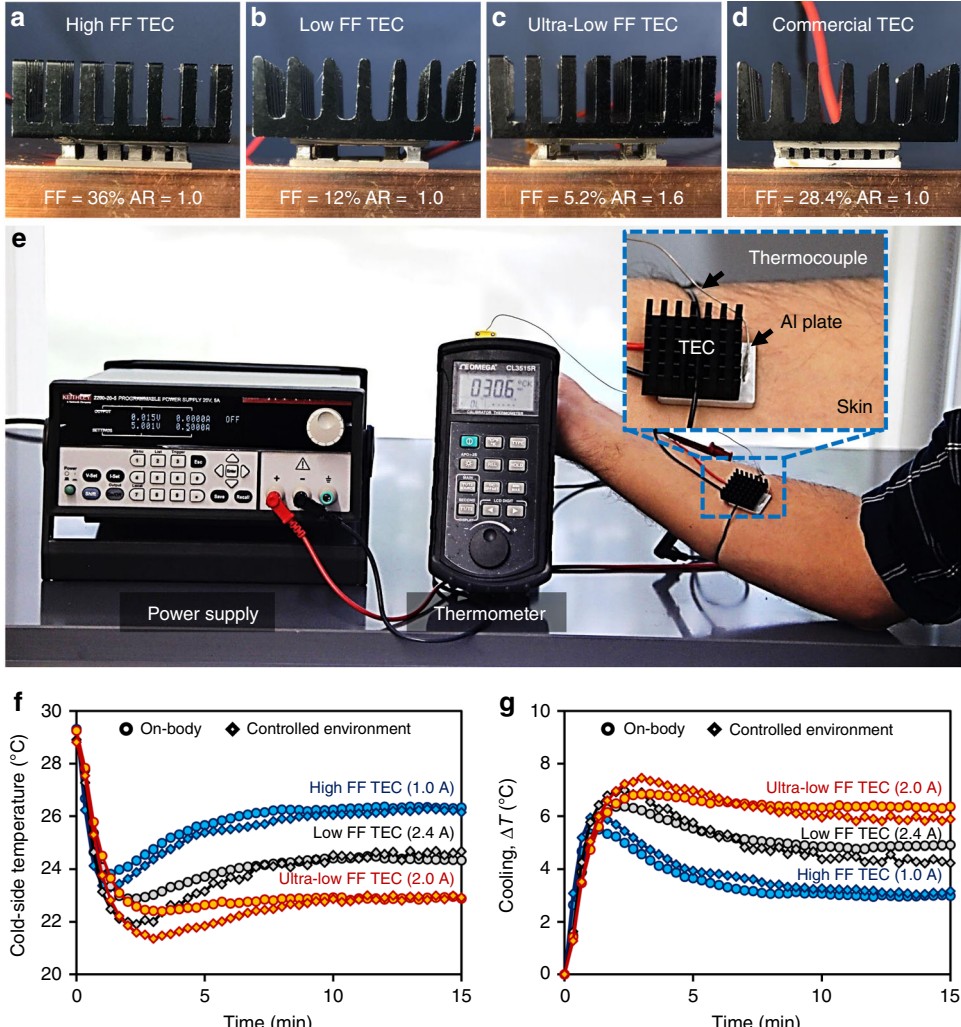

**Fig. 5** TEC modules and experimental set-up. Fabricated TEC modules: **a** high-fill factor (FF = 36% and AR = 1.0), **b** low-fill factor (FF = 12% and AR = 1.0), **c** ultra-low fill factor (FF = 5.2% and AR = 1.6), and **d** a commercial TEC module (FF = 28.4% and AR = 1.0). **e** Experimental set-up used to characterize the TEC modules on human body. **f**, **g** Transient study performed on different FF TECs under controlled environment and on human body. The transient minima are lower than steady state cold-side temperature for all TECs. The ultra-low FF TEC generates least cold-side temperature and thus maximum cooling followed by low-FF TEC and high-FF TEC

on cold-side of TECs was noted to be 30.6 °C, which is also approximately the skin temperature of human forearm under the normal ambient temperature of 22 °C. As the electric current is passed, the cold-side temperature reduces until it reaches the lowest value. Further increase in current results in increase in the cold-side temperature. For the high-FF TEC module, the least cold-side temperature was found to be 25.6 °C at $I \sim 1.0$ A. The low-FF TEC module resulted in the least cold-side temperature of 22.6 °C at $I \sim 2.5$ A, whereas the ultra-low FF TEC module generated the least cold-side temperature of 22.4 °C at $I \sim 2.1$ A. It is interesting to note that the optimal current is high when the internal electrical resistance of TEC module is low. Large electrical resistance of TEC modules results in high-joule heating, which negatively affects TEC cooling. Comparing with the initial temperature, the high-FF TEC module was found to result in maximum cooling of 5.0 °C, whereas the low and the ultra-low FF TEC modules resulted in maximum cooling of 8.0 and 8.2 °C, respectively. We can note that the ultra-low FF TEC module results in 1.6 times higher cooling than the high-FF TEC module.

In order to emphasize the cost-benefit analysis, cooling per unit input electrical power and cooling per unit volume of TE materials are illustrated in Fig. 7. It can be noted that at a fixed

current, cooling per unit input electric power is highest for the low-FF TEC followed by the ultra-low FF TEC and the high-FF TEC. For instance, at 1.1 A, the average cooling per unit input electric power is 8.9 °C W$^{-1}$ for the high-FF TEC, 17.6 °C W$^{-1}$ for the low-FF TEC, and 15.8 °C W$^{-1}$ for the ultra-low FF TEC. This indicates that the low FF TEC requires two times less electrical power than the high-FF TEC and 1.1 times less electrical power than the ultra-low FF TEC to generate same amount of cooling. In addition, since TE materials, especially the nanostructured materials, are expensive; therefore, the TE material volume has direct impact on the cost of TEC modules. It can be noted from Fig. 7 that, at optimal current, the cooling over TE material volume is 0.04 °C mm$^{-3}$ for the high-FF TEC, 0.16 °C mm$^{-3}$ for the low-FF TEC, and 0.37 °C mm$^{-3}$ for the ultra-low FF TEC. This indicates that the ultra-low FF TEC requires about nine times less TE materials than the high-FF TEC and 2.3 times less TE materials than the low-FF TEC to generate same amount of cooling.

It is important to note that higher cooling by the low and ultra-low FF TECs does not naturally result in a better thermal comfort. In fact, a large temperature drop on human skin can be quite uncomfortable; therefore, for practical purposes, users need

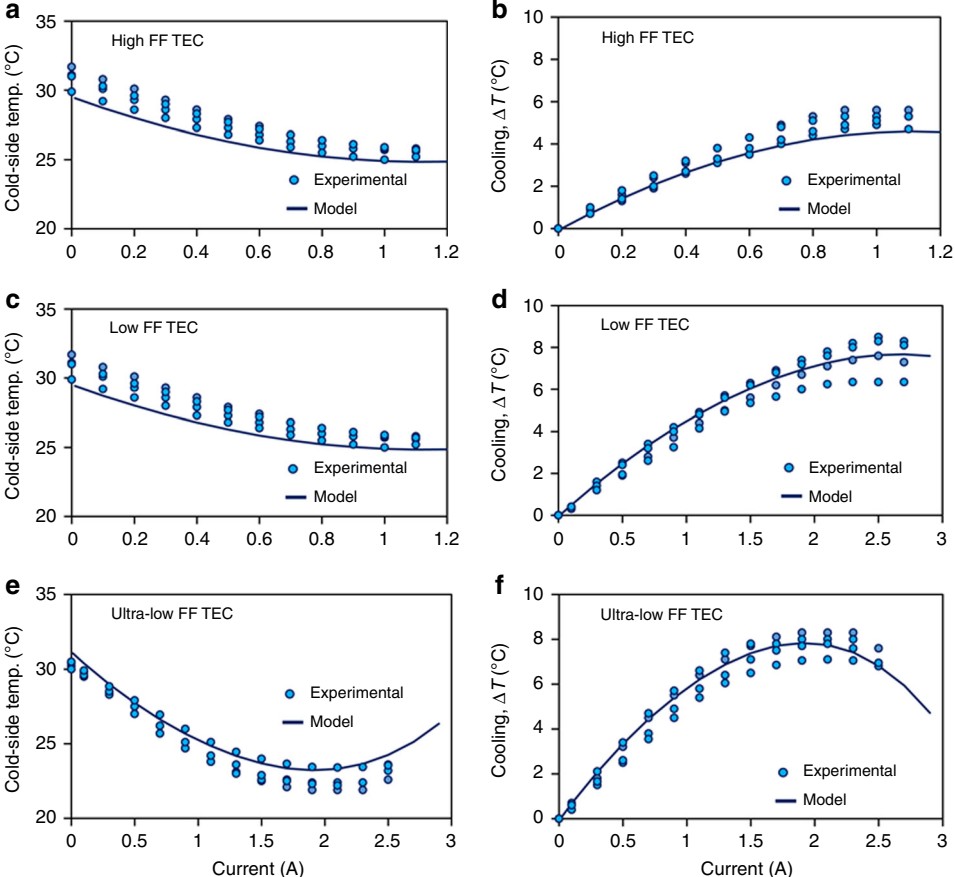

**Fig. 6** Cold-side temperature and cooling versus electric current. **a**, **b** High-FF TEC, **c**, **d** low-FF TEC, **e**, **f** Ultra-low FF TEC. Circles illustrate the experimental data, whereas solid lines show the modeling results (obtained with $h_{skin} = 100 \, W \, m^{-2} \, K^{-1}$). The ultra-low FF TEC module results in 1.6 times higher cooling than the high-FF TEC module

to be provided with a temperature controller to control the cooling based on thermal preferences. None-the-less, since the low- and ultra-low FF TECs proposed in this study have capability to generate a larger temperature drop than the high-FF TEC, they are expected to be more effective in hot-weather conditions. Figure 8a–c depicts the cold-side temperature of the different TECs deployed on human body under different ambient temperatures. It can be noted that the initial skin temperature is higher at higher ambient temperature, indicating the fact that the human body temperature increases with increase in ambient temperature. The average initial skin temperature was noted to be 30.6 °C under ambient temperature of 22 °C, 32.1 °C under ambient temperature of 26 °C, and 34.6 °C under ambient temperature of 32 °C. In Fig. 8a, when ambient temperature is 22 °C, the high-FF TEC generates minimum cold-side tempera-ture of 25.2 °C, which increases to 27.7 °C under ambient temperature of 26 °C, and 31.7 °C under ambient temperature of 32 °C. In Fig. 8b, the low-FF TEC generates minimum cold-side temperature of 22.4 °C under ambient temperature of 22 °C, 25 °C under ambient temperature of 26 °C, and 29 °C under ambient temperature of 32 °C. On the other hand, in Fig. 8c, the ultra-low FF TEC module generates minimum cold-side temperature of 22.3 °C under ambient temperature of 22 °C, 24.9 °C under ambient temperature of 26 °C, and 28.7 °C under ambient temperature of 32 °C. It is important to note that when ambient temperature is 32 °C, the high-FF TEC is unable to cool the skin temperature to 30.5 °C, which is typically the skin temperature in normal ambient condition of 22 °C. This implies that in hot climate the high-FF TEC is ineffective in cooling the

human body. The low-FF and the ultra-low FF TECs, on the other hand, can be observed to cool the human skin below 29 °C, when ambient temperature is 32 °C, emphasizing the fact that these modules are quite effective even in extreme climate.

Figure 8d–f illustrates the cooling flux for different TECs deployed on human body under different ambient temperatures. It can be noted that when the applied electric current is zero, depending upon the ambient temperature, the heat flux from human skin varies in the range of 15–50 mW cm$^{-2}$. The cooling flux increases with increase in electric current, but it decreases with increase in ambient temperature. The peak cooling flux can be observed in the range of 85–110 mW cm$^{-2}$ for the high-FF TEC, 120–140 mW cm$^{-2}$ for the low-FF TEC, and 115–130 mW cm$^{-2}$ for the ultra-low FF TEC. It has been suggested that if a localized thermal management system is able to remove 23 W of heat from human body, the cooling setpoint of household heating, ventilation, and air conditioning system can be increased by 2 °C, leading to considerable saving in energy consumption[57]. Considering the surface area of 1.8 m$^2$ for an average adult[57], it can be calculated that 23 W of body heat can be removed by covering 1.0–2.0% of the body surface with TECs (refer Supplementary Note 4, Supplementary Table 4, and Supplemen-tary Table 5 for more discussion).

Figure 9 compares the maximum cooling and cooling per unit volume of the TE materials for different TECs considered in this study including a commercial TEC deployed on human body under ambient temperature of 22 °C. The detailed experimental results for the commercial TEC are illustrated in Supplementary Fig. 5. It can be noted that the commercial TEC generates

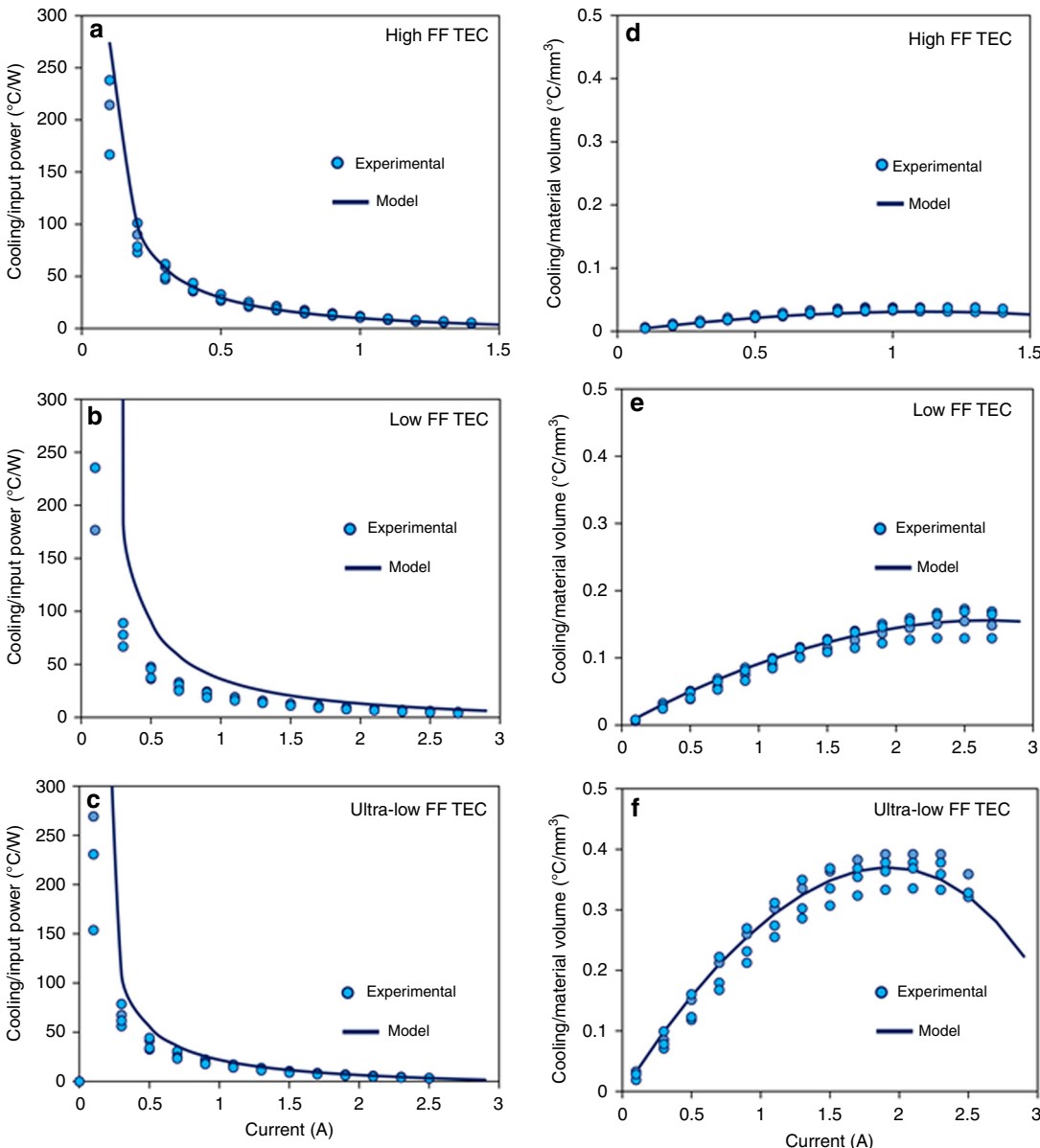

**Fig. 7** Cooling over electric power and cooling over material volume. **a**–**c** For a fixed electric current, cooling per unit input electric power is highest for the low-FF TEC followed by the ultra-low FF TEC and the high FF TEC. **d**–**f** Under optimal electric current, the cooling over TE material volume is highest for the ultra-low FF TEC, followed by the low-FF TEC and the high-FF TEC. Circles illustrate the experimental data, whereas solid lines show the modeling results

maximum cooling of 4.9 °C against 5.0 °C by the high-FF TEC, 8.0 °C by the low-FF TEC and 8.2 °C by the ultra-low FF TEC fabricated in this study. This indicates that our ultra-low FF TEC generates 1.7 times higher cooling than the commercial TEC. The maximum cooling per unit material volume is 0.08 °C/mm³ for the commercial TEC against 0.4 °C/mm³ for the ultra-low FF TEC, which is five times higher than that of the commercial TEC.

## Discussion

In summary, we demonstrated that the thermal resistance of the operating environment influences the TEC design that requires specific combination of material properties and module parameters. It was observed that when thermal resistance of the operating environment is small, TE material with higher Seebeck coefficient provides higher cooling capacity and TE material with higher electrical conductivity results in higher COP. However, as

the external thermal resistance increases, the effect of TE thermal conductivity becomes increasingly important. When heat source and sink resistances are very high, TE material with lower thermal conductivity generates greater cooling and higher COP. The heat source and sink resistances also have a very strong impact on the TEC module design. The TEC modules used in low thermally resistive environment should have high-FF (>20%), whereas TECs used in high-thermally resistive environment should have low-FF (<20%). The commercial TEC modules usually have FF greater than 25% and therefore, they cannot be considered as the best TECs for wearable devices. The TECs used in wearable devices should have FF less than 15% and AR of 1–2. Experiments revealed that TECs with FF of 36% and leg AR of one resulted in body cooling of 5.0 °C, whereas the TEC with FF of 5.2% and leg AR of 1.6 caused body cooling of 8.2 °C. Comparing against a commercial TEC module, the ultra-low FF TEC module was found to generate 1.7 times higher cooling. Also, cooling over

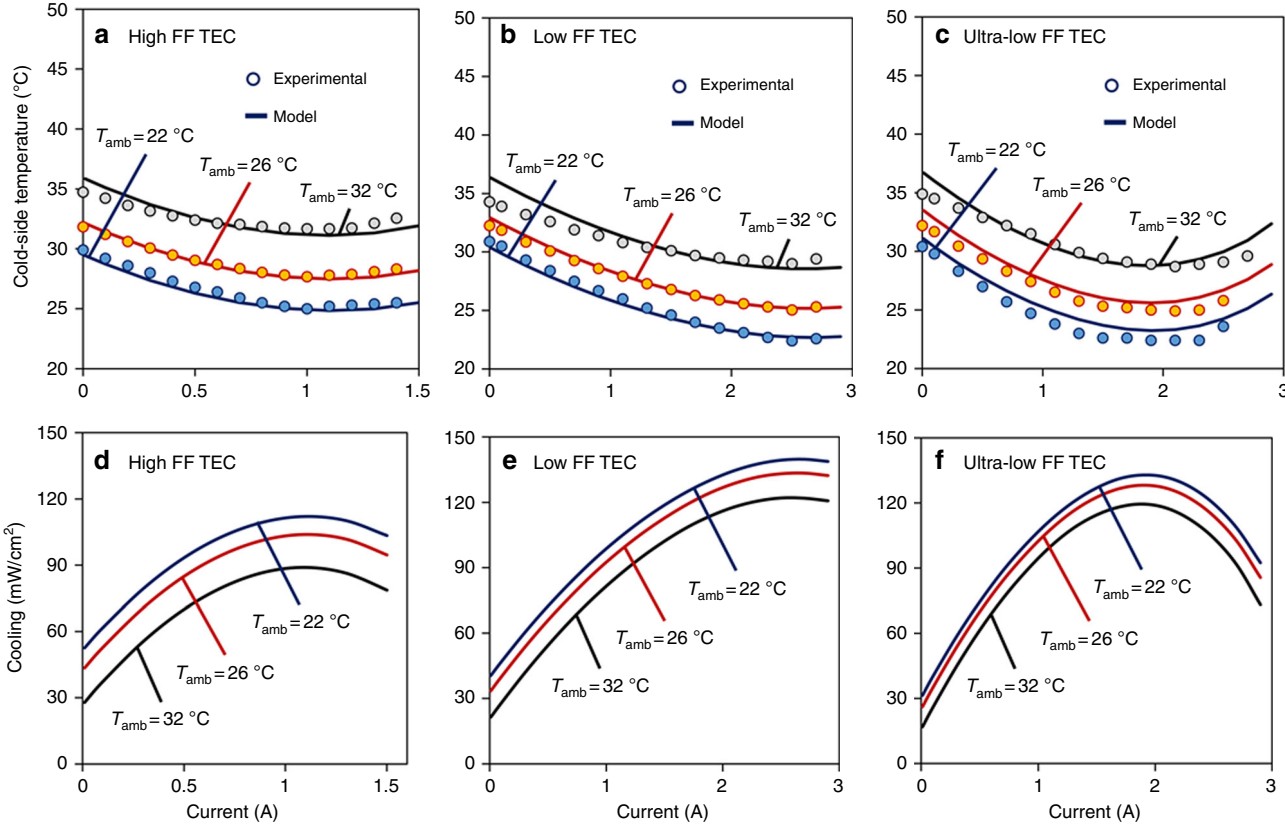

**Fig. 8** Cold-side temperature and cooling for TECs on human body. Increase in ambient temperature negatively impacts the cooling performance of the TECs. **a–c** In a hot climate (ambient temperature ~32 °C), the high-FF TEC is unable to cool the skin temperature to its normal temperature (~30.5 °C), whereas the low- and ultra-low FF TECs are effective even in extreme climate. **d–f** The numerical results illustrating cooling flux versus electric current for different FF TECs deployed on human body under different ambient temperatures. The cooling flux increases with increase in electric current, but it decreases with increases in ambient temperature. Circles illustrate the experimental data whereas solid lines show the modeling results

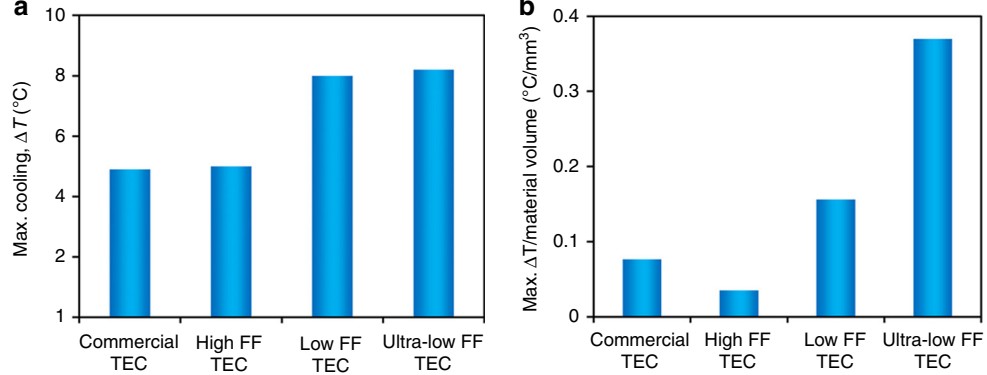

**Fig. 9** Performance comparison among commercial and fabricated TECs. **a** Maximum cooling (temperature drop from initial temperature). The ultra-low FF TEC provides 1.7 times higher cooling than the commercial TEC. **b** Maximum cooling over volume of TE materials. The cooling over material volume for the ultra-low FF TEC is five times higher than that of the commercial TEC

material volume for the ultra-low FF TEC was found to be five times higher than that of the commercial TEC.

## Methods

**Modeling**. Supplementary Fig. 6a schematically illustrates a TEC module placed between a heat source and a heat sink. A TEC consists of several p- and n-type legs connected electrically in series and thermally in parallel. When electric current is passed through a TEC, holes in p-type and electrons in n-type legs move in the direction of the current, which carry thermal energy thereby creating cold and hot sides of the TEC module. TE effect is a complex phenomenon as it is a combination of four different effects: Seebeck effect, Peltier effect, Thomson effect, and Joule

heating[58] (please refer Supplementary Note 1 for more details). An accurate TEC model, therefore, requires multi-physics coupled expressions. Building upon this fundamental understanding, a simplified one-dimensional energy equilibrium model is presented here, which will be used for explaining the key observations. The complex three-dimensional model, which is used to obtain the numerical results in this paper, is presented in the next section.

**Simplified one-dimensional energy equilibrium model**. The thermal resistances due to heat source, heat sink, and TEC constitute a thermal circuit as shown in Supplementary Fig. 6b. In the simplified one-dimensional model, it is assumed that the TEC is operating under steady state condition, material properties are

temperature-independent (ignoring Thomson effect), and the heat losses to the environment are negligible. Performing the energy balance on the cold and hot sides of TEC, it can be shown that[59]

$$Q_c + \frac{Q_j}{2} - Q_{pc} + Q_{TEC} = 0, \tag{4}$$

$$-Q_h + \frac{Q_j}{2} + Q_{ph} - Q_{TEC} = 0, \tag{5}$$

where $Q_c$ is the heat flow from the heat source (termed as cooling capacity of the TEC), $Q_h$ is the heat flow to the heat sink, $Q_{TEC}$ is heat flow through TEC because of the internal temperature difference, $Q_{pc}$ is Peltier effect generated cooling on the cold-side, $Q_{ph}$ is Peltier effect generated heating on the hot-side, and $\frac{Q_j}{2}$ is the contribution due to Joule heating. These terms can be mathematically expressed as[60]

$$Q_c = \frac{T_{source} - T_c}{R_{source}}, \tag{6}$$

$$Q_h = \frac{T_h - T_{sink}}{R_{sink}}, \tag{7}$$

$$Q_{TEC} = \frac{T_h - T_c}{R_{tTEC}}, \tag{8}$$

$$Q_{pc} = n\alpha T_c I, \tag{9}$$

$$Q_{ph} = n\alpha T_h I, \tag{10}$$

$$Q_j = I^2 R_{eTEC}, \tag{11}$$

where $T_{source}$ and $T_{sink}$ are the core temperatures of the heat source and sink, $R_{source}$ and $R_{sink}$ are the total thermal resistances of the heat source and sink, $T_h$ and $T_c$ are the temperatures of the hot- and cold-sides of the TEC, $R_{tTEC}$, and $R_{eTEC}$ are the internal thermal and electrical resistances of the TEC, $n$ is the total number of pairs of p- and n-type legs, and $\alpha$ is the net Seebeck coefficient given as $\alpha = \alpha_p - \alpha_n$, where $\alpha_p$ and $\alpha_n$ are the Seebeck coefficients of p- and n-type TE legs, respectively. Lastly, as shown in Supplementary Fig. 6c, $I$ is the current in the electrical circuit because of a current source. Using Eqs. (4)–(11), the cooling capacity ($Q_c$) of a TEC can be expressed as

$$Q_c = n\alpha T_c I - \frac{I^2 R_{eTEC}}{2} - \frac{T_h - T_c}{R_{tTEC}}. \tag{12}$$

$V_{se}$ shown in Supplementary Fig. 6c is the Seebeck voltage, which is given as[61]

$$V_{se} = n\alpha(T_h - T_c). \tag{13}$$

The input electrical power $P_{in}$ by the current source is given by[62]

$$P_{in} = Q_h - Q_c = I^2 R_{eTEC} + n\alpha I(T_h - T_c). \tag{14}$$

The COP of TEC is measured using relation

$$COP = \frac{Q_c}{P_{in}}. \tag{15}$$

It should be noted that cooling capacity ($Q_c$) and COP of TEC are a function of the internal temperature difference $\Delta T_i = T_h - T_c$, which is usually not known. When $R_{source}$ and $R_{sink}$ are negligible, it can be assumed that $T_c \approx T_{source}$ and $T_h \approx T_{sink}$. However, for the applications where heat source and/or sink resistances are considerably high, cooling capacity and COP are greatly affected by these external resistances.

The cooling capacity of TEC constitutes of three components (Eq. (12)): Peltier cooling ($Q_{pc}$), Joule heating ($Q_j$), and the internal heat flow ($Q_{TEC}$). In order to increase cooling capacity, $Q_{pc}$ should be increased, while $Q_j$ and $Q_{TEC}$ should be lowered. Increase in Seebeck coefficient ($\alpha$) increases $Q_{pc}$ (Eq. (9)), thus it has a positive effect on cooling capacity. Joule heat $Q_j$ is a function of TEC internal electrical resistance ($R_{eTEC}$), which is given as[63]

$$R_{eTEC} = n\frac{l}{a}\left(\frac{1}{\sigma_p} + \frac{1}{\sigma_n}\right), \tag{16}$$

where $n$ is the number of leg pairs, $l$ is the leg height, $a$ is the leg cross-sectional area, $\sigma_p$ and $\sigma_n$ are electrical conductivity of p- and n-type legs. Increase in electrical conductivity ($\sigma$) decreases $R_{eTEC}$ and thus reduces $Q_j$ (Eq. (11)). Hence, it also has a positive effect on cooling capacity. Likewise, the internal thermal resistances of a TE module is given as[63]

$$R_{tTEC} = \frac{1}{n}\frac{l}{a}\frac{1}{(\kappa_p + \kappa_p)}, \tag{17}$$

where $\kappa_p$ and $\kappa_p$ are thermal conductivity of p- and n-type legs. Lowering the value of material thermal conductivity ($\kappa$) increases $R_{tTEC}$, which reduces $Q_{TEC}$ (Eq. (8)) and thus enhances cooling capacity.

**Three-dimensional coupled multi-physics model.** The simplified one-dimensional model presented in the previous section is a decoupled model where thermal and electrical relations are solved separately. One-dimensional model explains the physics and provides fairly accurate results (error up to 10%)[59,64] when thermal gradient is small, material properties are temperature-independent, and contact resistances are small[65]. However, for a more robust analysis, a three-dimensional model is needed to account for temperature-dependent material properties and various types of losses. In order to obtain numerical results provided in this paper, the complex three-dimensional equations of TEity in steady state are used, which are given as[66]

$$\nabla(\kappa \nabla T) + \frac{J^2}{\sigma} - TJ.\left[\left(\frac{\partial \alpha}{\partial T}\right)\nabla T + (\nabla \alpha)_T\right] = 0, \tag{18}$$

$$\nabla.J = 0, \tag{19}$$

Current density vector $\mathbf{J}$ and heat flux vector $\mathbf{q}$ for TEC model in three dimensions, are given as[66]

$$\mathbf{J} = -\sigma(\nabla V + \alpha \nabla T), \tag{20}$$

$$\mathbf{q} = \alpha T\mathbf{J} - \kappa \nabla T. \tag{21}$$

where $V$ is the electrostatic potential, $T$ is the absolute temperature, and $\kappa$, $\sigma$, and $\alpha$ denote the temperature-dependent thermal conductivity, electrical conductivity, and Seebeck coefficient of the TE materials. Eqs. (18)–(21) are coupled and thus need to be solved numerically. In this study, finite element calculations were performed using the commercial code, ANSYS (Version 18.1).

**Module fabrication.** The TEC modules used this study are built using the commercial p- and n-type bismuth telluride materials. The high-FF TEC module contains 36 legs, whereas the low-FF and ultra-low FF TEC modules contain 12 legs. The leg cross-sectional area of the high-FF and the low-FF TEC modules is $1.6 \times 1.6$ mm$^2$, whereas the leg cross-sectional area of the ultra-low FF TEC module is $1.05 \times 1.05$ mm$^2$. The leg height of all the TEC modules is 1.6 mm. The copper electrodes and the aluminum nitride substrates have thickness of 0.14 and 0.64 mm. The base area of the TEC modules is $16 \times 16$ mm$^2$. The internal electrical resistance of the TEC modules at room temperature was measured using four probe method and was found to be 228, 75, and 185 m$\Omega$ for high-FF, low-FF, and ultra-low FF TEC modules, respectively.

**Measurements.** All the temperature readings in this study were obtained using K-type thermocouples, purchased from Omega Engineering. Keithley 2200-20-5 power supply was used to supply electric current to the TEC module during cooling measurements.

## Data availability

The data that support the findings of this study are available from the corresponding author upon reasonable request.

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

## Acknowledgements

The authors (R.A.K. and S.P.) gratefully acknowledge the financial support from DARPA MATRIX program (NETS). R.A.K. acknowledges the financial support from ICTAS Doctoral Scholars Program. A.N. would like to acknowledge the financial support through Army Research Office (ARO) through DARPA TE3 program. B.P. acknowledges the financial support from National Science Foundation.

## Author contributions

R.A.K. and B.P. conceived the idea. R.A.K. performed the simulations. A.N. and B.P. fabricated the TEC modules. R.A.K., B.P., and A.N. performed the experiments. M.S. contributed in the discussions of results. S.P. provided input on the implementation and supervised the overall research.

## Additional information

**Competing interests:** The authors declare no competing interests.

