## [Peer Review File · Nature Communications]

Reviewers' comments:

Reviewer #1 (Remarks to the Author):

This manuscript presented a study of TECs for the use in cooling of human body, through both theoretical modelling and experimental measurement. It sounds an interesting and valuable work. However, a few items may need further strengthening and emphasis: (1) mathematical equations that support the operation of the TECs cooling should be defined and presented; (2) model establishment process should be elaborated; (3) comparison between the modelling and experimental results should be made intensively; (4) optimised material and configuration should be sorted out and presented more clearly; and (5) Novelty and added value of the research should be stressed.

Reviewer #2 (Remarks to the Author):

This work investigated the combined effect of the thermal resistance and TE material properties on TEC performance. The manuscript showed that the optimized design achieved human skin cooling to 8.5°C below the ambient temperature, with a better cooling over material volume.

(1) First, as referred in the manuscript, the references [14, 22-24] have developed wearable wrist TEC devices for commercialization but there is no comprehensive review on current technologies for appropriate propositions of this work (e.g., additional online resource: <http://news.mit.edu/2017/personal-thermostat-startup-heats-commercialization-0927>).

(2) Page 4, it is stated that "the effect of external thermal resistances on TEC performances has not been well-examined". There are quite a number of researches focusing on the impact of thermal resistances on TEC performances. The authors should give a complete review on this topic and explain why it hasn't been well-examined. In addition, the Introduction part should be more concentrated on the points that would lead to this work's innovations instead of talking too much about well-known things. When referring other works, the manuscript should directly touch the most critical and relevant points, not just list the references there.

(3) Page 6, "Despite these discoveries, the performance of TE devices during real deployments has remained poor [58], indicating a strong influence of operating environment on the device performance." The work [58] referred here mainly talks about high demand of breakthrough in TE material development. Apparently it doesn't indicate the strong influence of operating environment on the device performance.

(4) Page 13, when designing on-body applications and personalized cooling TEC, what criteria should be the best to judge the quality of the TEC: temperature cooled, cooling heat flux, COP, or defined cooling/material volume and cooling/input power (Fig. S6)? Previous work [31-32] found human body emits thermal energy at the rate of ~ 25 mW/cm² into the ambient environment. Meanwhile, [59] presented the findings on thermal comfort of human body under personal thermal management with a conclusion that different body parts may require different temperatures. Is the greater temperature drop naturally producing better thermal comfort? Therefore, a well justified logic foundation is required to support the conclusions made from Fig. 6. Also, an explanation might be needed for the reason why switching from cooling/area (W/cm²) of Fig. 2 and Fig. 3(a,b) to cold-side temperature (°C) of Fig. 3(c,d), Fig. 4, and Fig. 6.

(5) Instead of directly applying the designed TECs to human body test, experiments under a controllable environment (e.g., fixed temperature or heat flux conditions) should be completed in order to more accurately examine the TEC performance. The heat transfer rate of the fin and heat flux from skin should be measured in the experiments, or at least the estimates should be verified.

Reviewer #3 (Remarks to the Author):

This paper focuses on the optimum design of Thermoelectric Coolers (TEC) operating under huge thermally resistive environment for localized wearable cooling applications. Nevertheless, the theoretical modeling and experiments conducted in this study were under the atmosphere temperature of 22 °C, which is not a typical hot condition requiring personal cooling. To demonstrate the cooling effect of TEC and potential energy saving in building HVAC systems through expansion of set-points as articulated in the introduction of this research, modeling and experiments should be carried out at an ambient temperature of above 26 °C (4 °F more than the normal set-points to achieve 20% saving in HVAC energy for cooling). Also, it is better to study the cooling effect of TEC at different ambient temperatures and show the influence of environment on its performance.

Besides, to maintain thermal comfort in hot environment, TEC should remove a certain amount of heat from human body. Based on the FOA of ARPAE's DELTA program, an additional 23W of heat removal is required if the up-bound of neutral set-points is increased by 4 °F. It is suggested to show heat removal the proposed TEC, given its covering area and weight. Furthermore, the large temperature reduction in localized skin area covered by the TEC can cause severe localized cold discomfort. It is advised to evaluate the local cold discomfort. Moreover, without an effective heat rejection in a hotter environment than the ambient temperature (22 °C) investigated in this study, it is advised to show whether the cooling effect is stable.

Reviewers' comments:

Reviewer #1 (Remarks to the Author):

This manuscript presented a study of TECs for the use in cooling of human body, through both theoretical modelling and experimental measurement. It sounds an interesting and valuable work. However, a few items may need further strengthening and emphasis:

(1) mathematical equations that support the operation of the TECs cooling should be defined and presented; (2) model establishment process should be elaborated;

Author's reply: Thanks a lot for the suggestions. The modeling and consecutive equations were initially provided in the Supplementary information. However, based on the reviewer's recommendation, we have now presented the important equations under the methods section in the revised manuscript. This section also elaborates the model development.

For convenience of reviewer, the modeling section in the revised manuscript is presented below.

Modeling

Fig. 10(a) schematically illustrates a TEC module placed between a heat source and a heat sink. A TEC consists of several p- and n-type legs connected electrically in series and thermally in parallel. When electric current is passed through a TEC, holes in p-type and electrons in n-type legs move in the direction of the current, which carry thermal energy thereby creating cold and hot sides of the TEC module. TE effect is a complex phenomenon as it is a combination of four different effects: Seebeck effect, Peltier effect, Thomson effect, and Joule heating⁶⁰ (please refer Supplementary information for more details). An accurate TEC model, therefore, requires multi-

physics coupled expressions. Building upon this fundamental understanding, a simplified one-dimensional energy equilibrium model is presented here, which will be used for explaining the key observations. The complex three-dimensional model, which is used to obtain the numerical results in this paper, is presented in the next section.

Simplified one-dimensional energy equilibrium model

The thermal resistances due to heat source, heat sink, and TEC constitute a thermal circuit as shown in Fig. 10(b). In the simplified one-dimensional model, it is assumed that the TEC is operating under steady state condition, material properties are temperature-independent (ignoring Thomson effect), and the heat losses to the environment are negligible. Performing the energy balance on the cold and hot sides of TEC, it can be shown that,⁶¹

$$Q_c + \frac{Q_j}{2} - Q_{pc} + Q_{TEC} = 0, \quad (4)$$

$$-Q_h + \frac{Q_j}{2} + Q_{ph} - Q_{TEC} = 0, \quad (5)$$

where Q_c is the heat flow from the heat source (termed as cooling capacity of the TEC), Q_h is the heat flow to the heat sink, Q_{TEC} is heat flow through TEC because of the internal temperature difference, Q_{pc} is Peltier effect generated cooling on the cold-side, Q_{ph} is Peltier effect generated heating on the hot-side, and $\frac{Q_j}{2}$ is the contribution due to Joule heating. These terms can be mathematically expressed as⁶²:

$$Q_c = \frac{T_{source} - T_c}{R_{source}}, \quad (6)$$

$$Q_h = \frac{T_h - T_{sink}}{R_{sink}}, \quad (7)$$

$$Q_{\text{TEC}} = \frac{T_h - T_c}{R_{t\text{TEC}}}, \quad (8)$$

$$Q_{\text{pc}} = n\alpha T_c I, \quad (9)$$

$$Q_{\text{ph}} = n\alpha T_h I, \quad (10)$$

$$Q_j = I^2 R_{e\text{TEC}}, \quad (11)$$

where T_{source} and T_{sink} are the core temperatures of the heat source and sink, R_{source} and R_{sink} are the total thermal resistances of the heat source and sink, T_h and T_c are the temperatures of the hot- and cold-sides of the TEC, $R_{t\text{TEC}}$ and $R_{e\text{TEC}}$ are the internal thermal and electrical resistances of the TEC, n is the total number of pairs of p- and n-type legs, and α is the net Seebeck coefficient given as $\alpha = \alpha_p - \alpha_n$, where α_p and α_n are the Seebeck coefficients of p- and n-type TE legs, respectively. Lastly, as shown in Fig. 10(c), I is the current in the electrical circuit because of a current source. Using equations (4)-(11), the cooling capacity (Q_c) of a TEC can be expressed as:

$$Q_c = n\alpha T_c I - \frac{I^2 R_{e\text{TEC}}}{2} - \frac{T_h - T_c}{R_{t\text{TEC}}}. \quad (12)$$

V_{se} shown in Fig. 10(c) is the Seebeck voltage, which is given as⁶³:

$$V_{\text{se}} = n\alpha(T_h - T_c) \quad (13)$$

The input electrical power P_{in} by the current source is given by⁶⁴:

$$P_{\text{in}} = Q_h - Q_c = I^2 R_{e\text{TEC}} + n\alpha I(T_h - T_c) \quad (14)$$

The coefficient of performance (COP) of TEC is measured using relation:

$$\text{COP} = \frac{Q_c}{P_{\text{in}}} \quad (15)$$

It should be noted that cooling capacity (Q_c) and COP of TEC are a function of the internal temperature difference $\Delta T_i = T_h - T_c$, which is usually not known. When R_{source} and R_{sink} are negligible, it can be assumed that $T_c \approx T_{source}$ and $T_h \approx T_{sink}$. However, for the applications where heat source and/or sink resistances are considerably high, cooling capacity and COP are greatly affected by these external resistances.

The cooling capacity of TEC constitutes of three components (Equation (12)): Peltier cooling (Q_{pc}), Joule heating (Q_j), and the internal heat flow (Q_{TEC}). In order to increase cooling capacity, Q_{pc} should be increased, while Q_j and Q_{TEC} should be lowered. Increase in Seebeck coefficient (α) increases Q_{pc} (Equation (9)), thus it has a positive effect on cooling capacity. Joule heat Q_j is a function of TEC internal electrical resistance (R_{eTEC}), which is given as⁶⁵:

$$R_{eTEC} = n \frac{l}{a} \left(\frac{1}{\sigma_p} + \frac{1}{\sigma_n} \right) \quad (16)$$

where n is the number of leg pairs, l is the leg height, a is the leg cross-sectional area, σ_p and σ_n are electrical conductivity of p- and n-type legs. Increase in electrical conductivity (σ) decreases R_{eTEC} and thus reduces Q_j (Equation (11)). Hence, it also has a positive effect on cooling capacity. Likewise, the internal thermal resistances of a TE module is given as⁶⁵:

$$R_{tTEC} = \frac{1}{n} \frac{l}{a} \frac{1}{(\kappa_p + \kappa_n)} \quad (17)$$

where κ_p and κ_n are thermal conductivity of p- and n-type legs. Lowering the value of material thermal conductivity (κ) increases R_{tTEC} , which reduces Q_{TEC} (Equation 8) and thus enhances cooling capacity.

Fig. 10. Simplified one-dimensional energy equilibrium model of a thermoelectric cooler (TEC). (a) Schematic illustration of the structure of a TEC, which consists of several p- and n-type legs connected electrically in series and thermally in parallel. Heat source and heat sink on the two sides of TEC can create a thermally resistive environment for TEC, affecting its design and performance. (b) One-dimensional thermal circuit illustrating the heat flow from heat source to heat sink via TEC. Performing energy balance on hot- and cold-sides provides mathematical relations that can be solved to obtain cooling capacity and the coefficient of performance (COP). (c) The electrical-circuit of TEC depicting the Seebeck voltage and internal electrical resistance.

Three-dimensional coupled multi-physics model

The simplified one-dimensional model presented in the previous section is a decoupled model where thermal and electrical relations are solved separately. One-dimensional model explains the physics and provides fairly accurate results (error up to 10%)^{61,66} when thermal gradient is small, material properties are temperature-independent, and contact resistances are small⁶⁷. However, for a more robust analysis, a three-dimensional model is needed to account for temperature

dependent material properties and various types of losses. In order to obtain numerical results provided in this paper, the complex three-dimensional equations of thermoelectricity in steady state are used, which are given as ⁶⁸:

$$\nabla(\kappa\nabla T) + \frac{J^2}{\sigma} - T\mathbf{J} \cdot \left[\left(\frac{\partial\alpha}{\partial T} \right) \nabla T + (\nabla\alpha)_T \right] = 0, \quad (18)$$

$$\nabla \cdot \mathbf{J} = 0, \quad (19)$$

Current density vector \mathbf{J} and heat flux vector \mathbf{q} for TEC model in three dimensions, are given as⁶⁸:

$$\mathbf{J} = -\sigma(\nabla V + \alpha\nabla T), \quad (20)$$

$$\mathbf{q} = \alpha T\mathbf{J} - \kappa\nabla T. \quad (21)$$

where V is the electrostatic potential, T is the absolute temperature, and κ , σ , and α denote the temperature dependent thermal conductivity, electrical conductivity, and Seebeck coefficient of the TE materials. Equations (18)-(21) are coupled and thus need to be solved numerically.

(3) comparison between the modelling and experimental results should be made intensively;

Author's reply: Thanks for the suggestion. We have added following figures in the revised manuscript where experimental and modeling results are compared to validate the numerical model. Figure 6 compares experimental and numerical results for cold-side temperature and cooling (ΔT) versus electric current for different fill factor TEC modules deployed on human body under ambient temperature of 22°C. In Figure 7, numerical and experimental data for cooling over thermoelectric material volume and cooling over input electrical power have been

compared for different fill factor TEC modules. Figure 8(a)-(c) depict experimental and numerical results for cold-side temperature versus electric current for different fill factor TEC modules deployed under different ambient temperatures (22°C, 26°C, and 32°C). The numerical results are in close agreement with the experimental results. Minor differences in the modeling and experimental results for cooling/input power at small current are due to external electrical resistances, such as connecting wires, connectors, etc., that were not accounted in the numerical model.

The figures included in the revised manuscript are presented below.

Fig. 6. Cold-side temperature and cooling (temperature drop from initial temperature) versus electric current for the different FF TEC modules. (a) and (b) High FF TEC, (c) and (d) low FF TEC, (e) and (f) Ultra-low FF TEC. Circles illustrate the experimental data whereas solid lines show the modeling results (obtained with $h_{\text{skin}} = 100 \text{ W/m}^2\text{-K}$). The ultra-low FF TEC module results in 1.6 times higher cooling than the high FF TEC module.

Fig. 7. Cooling per unit input electric power and cooling over TE material volume for different TECs. (a)-(c) For a fixed electric current, cooling per unit input electric power is highest for the low FF TEC followed by the ultra-low FF TEC and the high FF TEC. (d)-(f) Under optimal electric current, the cooling over TE material volume is highest for the ultra-low

FF TEC, followed by the low FF TEC and the high FF TEC. Circles illustrate the experimental data whereas solid lines show the modeling results.

Fig. 8(a)-(c). Cold-side temperature for different TECs deployed on human body under different ambient temperatures. Increase in ambient temperature negatively impacts the cooling performance of the TECs. In a hot climate (ambient temperature $\sim 32^{\circ}\text{C}$), the high FF TEC is unable to cool the skin temperature to its normal temperature ($\sim 30.5^{\circ}\text{C}$), whereas the low and ultra-low FF TECs are effective even in extreme climate. Circles illustrate the experimental data whereas solid lines show the modeling results.

(4) optimised material and configuration should be sorted out and presented more clearly;

Author's reply: Thanks again for the suggestions. Considering the constraint on the word count for the journal, optimized TE material and TEC module configuration are briefly described in the revised manuscript. However, based on the reviewer's recommendation, we have added detailed discussion on this topic in the revised Supplementary information.

The information below has been added in the revised Supplementary information.

It can be noted that due to high thermal resistivity of the human skin and the ambient air along with the size constraint of the heat sink for on-body applications, wearable TECs operate in extremely high thermally resistive environment. This is evident from the heat transfer coefficient of human skin and heat sink, which is less than $100 \text{ W/m}^2\text{-K}$. The TE material composition and module design for on-body applications are, therefore, quite different from the typical commercial modules. The TEC modules are typically deployed in applications where external (heat source and heat sink) thermal resistances are low. In these cases, TE materials with high zT and large Seebeck coefficient and electrical conductivity are desired. However, in applications where heat source and heat sink resistances are large, such as human body, the TE material with low thermal conductivity is required. In this study, in order to minimize the cost, we have utilized commercial bismuth telluride TE materials for fabricating the different fill factor TEC modules. Thermal conductivity of the TE materials used in the study is $\sim 1.25 \text{ W/m-K}$ for p-type and $\sim 1.5 \text{ W/m-K}$ for n-type at 30°C . Seebeck coefficient at 30°C is $\sim 220 \mu\text{V/m}$ and electrical resistivity at 30°C is $\sim 10.8 \mu\Omega\text{-m}$ for both p- and n-type materials. Collectively, the material zT at 30°C is found to be 1.1 for p-type and 0.87 for n-type TE materials. The key dimensional features of TEC modules used in this study are provided in Table S3. The high FF TEC module contains 18 leg pairs, whereas the low FF TEC and the ultra-low FF TEC modules contains 6 leg pairs. The dimensions of p- and n-type legs are 1.6 mm (length) \times 1.6 mm (width) \times 1.6 mm (height) for the high and low FF TECs, whereas the ultra-low FF TEC contains legs of dimensions: 1.05 mm (length) \times 1.05 mm (width) \times 1.6 mm (height). The base area of all the TEC modules is $16 \text{ mm} \times 16 \text{ mm}$. This indicates that the high FF TEC has fill factor of 36% and leg aspect ratio of 1.0, the low FF TEC has fill factor of 12% and leg aspect ratio of 1.0, and the ultra-low FF TEC has fill factor of 5.2% and leg aspect ratio of 1.6. The internal electrical

resistance measured using four probe method was found to be 228 mΩ, 75 mΩ, and 185 mΩ for high FF, low FF, and ultra-low FF TEC modules, respectively.

Table S3. The key features of the fabricated TEC modules

Description	High FF	Low FF	Ultra-low FF
Module area (mm ²)	16 × 16	16 × 16	16 × 16
Leg height (mm)	1.6	1.6	1.6
Leg base area (mm ²)	1.6 x 1.6	1.6 x 1.6	1.05 x 1.05
Fill factor	36%	12%	5.2%
Electrode material	Cu	Cu	Cu
Electrode thickness (mm)	0.14	0.14	0.14
Substrate material	AlN	AlN	AlN
Substrate thickness (mm)	0.64	0.64	0.64
TEC electrical resistance (mΩ) at room temperature	228	75	185

and (5) Novelty and added value of the research should be stressed.

Authors' reply: Thanks again for the suggestions. We have thoroughly modified the introduction section, elaborating the novelty and importance of this work.

Following information has been added in the revised introduction.

Prior studies have reported a very strong influence of external thermal resistance on the performance of TECs ³³⁻³⁵ and suggested that the optimal TEC should have internal thermal

resistance in the range of 40-70% of the total thermal resistance of the system ³⁶. However, most of these studies were focused on electronic cooling using TECs. For on-body applications, the research in the field of thermoelectric, so far, has been focused towards thermal energy harvesting from body heat ³⁷⁻⁴³. The thermoelectric modules with fill factor less than 20% have been recommended for on-body applications, which brings challenges in terms of mechanical stability ⁴⁴. Thermoelectric modules for on-body cooling applications have not been much explored in the literature. A flexible thermoelectric system recently reported for human body demonstrated a temperature drop of 4°C, indicating the feasibility of using TECs to control the temperature of the human body ⁴⁵. While there are several commercial TEC products in the market, their material composition and design are proprietary. In this paper, we provide fundamental insight on designing TECs that can be deployed in different thermally resistive environments including the on-body conditions. The combined effect of each TE material property and the resistive condition on TEC performance is described in conjunction with TEC design parameters (leg dimensions, fill factor, aspect ratio, etc.). Using numerical and experimental studies, different TEC design architectures are investigated specifically for on-body applications under different ambient conditions and an optimal design for wearable TECs is obtained. The ultra-low fill fraction module proposed in this paper exhibits 170% higher cooling, while utilizing five times less TE materials than the commercial TEC.

Reviewer #2 (Remarks to the Author):

This work investigated the combined effect of the thermal resistance and TE material properties on TEC performance. The manuscript showed that the optimized design achieved human skin cooling to 8.5°C below the ambient temperature, with a better cooling over material volume.

(1) First, as referred in the manuscript, the references [14, 22-24] have developed wearable wrist TEC devices for commercialization but there is no comprehensive review on current technologies for appropriate propositions of this work (e.g., additional online resource: <http://news.mit.edu/2017/personal-thermostat-startup-heats-commercialization-0927>).

Authors' reply: Our apologies for the confusion. Please note that the material composition and the module design for commercial modules are propriety. Research on thermoelectric modules for on-body applications, thus far, has been primarily focused towards thermal energy harvesting from body heat. These studies have demonstrated that commercial modules are not optimal for on-body application. We have greatly modified the introduction section of the manuscript and have added a comprehensive review emphasizing the need of this study.

A portion of the revised introduction is given below.

Prior studies have reported a very strong influence of external thermal resistance on the performance of TECs³³⁻³⁵ and suggested that the optimal TEC should have internal thermal resistance in the range of 40-70% of the total thermal resistance of the system³⁶. However, most of these studies were focused on electronic cooling using TECs. For on-body applications, the research in the field of thermoelectric, so far, has been focused towards thermal energy harvesting from body heat³⁷⁻⁴³. The thermoelectric modules with fill factor less than 20% have been recommended for on-body applications, which brings challenges in terms of mechanical stability⁴⁴. Thermoelectric modules for on-body cooling applications have not been much explored in the literature. A flexible thermoelectric system recently reported for human body demonstrated a temperature drop of 4°C, indicating the feasibility of using TECs to control the temperature of the human body⁴⁵. While there are several commercial TEC products in the market, their material composition and design are proprietary. In this paper, we provide

fundamental insight on designing TECs that can be deployed in different thermally resistive environments including the on-body conditions. The combined effect of each TE material property and the resistive condition on TEC performance is described in conjunction with TEC design parameters (leg dimensions, fill factor, aspect ratio, etc.). Using numerical and experimental studies, different TEC design architectures are investigated specifically for on-body applications under different ambient conditions and an optimal design for wearable TECs is obtained. The ultra-low fill fraction module proposed in this paper exhibits 170% higher cooling, while utilizing five times less TE materials than the commercial TEC.

(2) Page 4, it is stated that “the effect of external thermal resistances on TEC performances has not been well-examined”. There are quite amount of researches focusing on the impact of thermal resistances on TEC performances. The authors should give a complete review on this topic and explain why it hasn’t been well-examined. In addition, the Introduction part should be more concentrated on the points that would lead to this work’s innovations instead of talking too much about well-known things. When referring other works, the manuscript should directly touch the most critical and relevant points, not just list the references there.

Authors’ reply: Thanks a lot for the suggestions. There are few studies reported in literature that have investigated the effect of external thermal resistances on the design and performance of TECs; however, these studies are focused towards electronic cooling. Due to high thermal resistivity of the human skin and the ambient air along with the size constraint of the heat sink for on-body applications, wearable TECs operate in extremely high thermally resistive environment. None-the-less, based on the reviewer’s recommendations we have modified the introduction section in the revised manuscript.

The portion of the revised introduction is given below.

Prior studies have reported a very strong influence of external thermal resistance on the performance of TECs³³⁻³⁵ and suggested that the optimal TEC should have internal thermal resistance in the range of 40-70% of the total thermal resistance of the system³⁶. However, most of these studies were focused on electronic cooling using TECs. For on-body applications, the research in the field of thermoelectric, so far, has been focused towards thermal energy harvesting from body heat³⁷⁻⁴³. The thermoelectric modules with fill factor less than 20% have been recommended for on-body applications, which brings challenges in terms of mechanical stability⁴⁴. Thermoelectric modules for on-body cooling applications have not been much explored in the literature. A flexible thermoelectric system recently reported for human body demonstrated a temperature drop of 4°C, indicating the feasibility of using TECs to control the temperature of the human body⁴⁵. While there are several commercial TEC products in the market, their material composition and design are proprietary. In this paper, we provide fundamental insight on designing TECs that can be deployed in different thermally resistive environments including the on-body conditions. The combined effect of each TE material property and the resistive condition on TEC performance is described in conjunction with TEC design parameters (leg dimensions, fill factor, aspect ratio, etc.). Using numerical and experimental studies, different TEC design architectures are investigated specifically for on-body applications under different ambient conditions and an optimal design for wearable TECs is obtained. The ultra-low fill fraction module proposed in this paper exhibits 170% higher cooling, while utilizing five times less TE materials than the commercial TEC.

(3) Page 6, “Despite these discoveries, the performance of TE devices during real deployments has remained poor [58], indicating a strong influence of operating environment on the device performance.” The work [58] referred here mainly talks about high demand of breakthrough in TE material development. Apparently it doesn’t indicate the strong influence of operating environment on the device performance.

Authors’ reply: Apology for the error. Reference has been corrected. The correct references are shown below.

[1] Lu, X., Zhao, D., Ma, T., Wang, Q., Fan, J., & Yang, R. (2018). Thermal resistance matching for thermoelectric cooling systems. Energy Conversion and Management, 169, 186-193.

[2] LeBlanc, S. (2014). Thermoelectric generators: Linking material properties and systems engineering for waste heat recovery applications. Sustainable Materials and Technologies, 1, 26-35.

(4) Page 13, when designing on-body applications and personalized cooling TEC, what criteria should be the best to judge the quality of the TEC: temperature cooled, cooling heat flux, COP, or defined cooling/material volume and cooling/input power (Fig. S6)? Previous work [31-32] found human body emits thermal energy at the rate of ~ 25 mW/cm² into the ambient environment. Meanwhile, [59] presented the findings on thermal comfort of human body under personal thermal management with a conclusion that different body parts may require different temperatures. Is the greater temperature drop naturally producing better thermal comfort? Therefore, a well justified logic foundation is required to support the conclusions made from Fig.

6. Also, an explanation might be needed for the reason why switching from cooling/area (W/cm^2) of Fig. 2 and Fig. 3(a,b) to cold-side temperature ($^{\circ}\text{C}$) of Fig. 3(c,d), Fig. 4, and Fig. 6.

Authors' reply: Thanks again. Cooling flux (W/cm^2) and coefficient of performance (cooling per unit input power) are the two most traditional metrics used in the literature to characterize TECs. The technical datasheet of commercial TECs usually also provides temperature drop (ΔT) at different heat loads. For body cooling, temperature drop can be considered a more appropriate criterion to determine the effectiveness of TECs as the temperature change can be easily perceived by the human body. However, in order to evaluate the electrical power needed to achieve the required temperature drop, a modified coefficient of performance (temperature drop per unit input electrical power) has been also reported in this study. Lastly, considering the human comfort and economic feasibility, weight, volume, and cost of the cooling devices are equally important. Therefore, we have reported cooling over material volume for different TECs studied in this paper.

We have added following explanation in the paper.

It should be noted that traditionally cooling capacity (W/m^2) and coefficient of performance (COP) are used in the literature to characterize TECs. For body cooling, however, cold-side temperature and temperature drop (ΔT) can be considered a more appropriate criterion to judge the effectiveness of TECs as temperature change can be easily perceived by the human body. Therefore, in the remainder of paper, cold-side temperature and temperature drop (ΔT) are reported in order to compare different TECs considered in this study. In addition, in order to evaluate the electrical power required to achieve the maximum temperature drop, a modified coefficient of performance defined as temperature drop over unit input electrical power is also calculated. Lastly, considering the human comfort and economic feasibility, weight, volume, and

cost of the cooling devices are equally important. Therefore, we have also reported cooling over material volume for different kinds of TECs studied in this paper.

The greater temperature drop does not naturally produce better thermal comfort. In fact, a very large temperature drop could be quite uncomfortable. For practical purposes, therefore, the users need to be provided with a temperature controller to control the temperature drop as per the desired comfort level. This study reveals that the low and ultra-low fill factor TEC modules have capability to produce a larger temperature drop than the high fill factor and commercial modules. This implies that the low fill factor TEC modules can be more effective in hot-weather conditions. Figure 8(a)-(c) in the revised manuscript illustrate that when ambient temperature is 32°C, the high fill factor TEC is unable to cool the human skin to 30.5°C, which is typically the skin temperature in normal ambient condition of 22°C. This infers that in hot climate, the high fill factor TEC is ineffective in cooling the human body. The low and ultra-low fill factor TEC modules, on the other hand, can be observed to cool the human skin below 29°C, when ambient temperature is 32°C, emphasizing the fact that these modules are quite effective even in extreme climate.

We have added following discussion in the revised manuscript.

It is important to note that higher cooling by the low and ultra-low FF TECs does not naturally result in a better thermal comfort. In fact, a large temperature drop on human skin can be quite uncomfortable; therefore, for practical purposes, users need to be provided with a temperature controller to control the cooling based on thermal preferences. None-the-less, since the low and ultra-low FF TECs proposed in this study have capability to generate a larger temperature drop

than the high FF TEC, they are expected to be more effective in hot-weather conditions. Fig. 8 (a)-(c) depict the cold-side temperature of the different TECs deployed on human body under different ambient temperatures. It can be noted that the initial skin temperature is higher at higher ambient temperature, indicating the fact that the human body temperature increases with increase in ambient temperature. The average initial skin temperature was noted to be 30.6°C under ambient temperature of 22°C, 32.1°C under ambient temperature of 26°C, and 34.6°C under ambient temperature of 32°C. In Fig. 8(a), when ambient temperature is 22°C, the high FF TEC generates minimum cold-side temperature of 25.2°C, which increases to 27.7°C under ambient temperature of 26°C, and 31.7°C under ambient temperature of 32°C. In Fig. 8(b), the low FF TEC generates minimum cold-side temperature of 22.4°C under ambient temperature of 22°C, 25°C under ambient temperature of 26°C, and 29°C under ambient temperature of 32°C. On the other hand, in Fig. 8(c), the ultra-low FF TEC module generates minimum cold-side temperature of 22.3°C under ambient temperature of 22°C, 24.9°C under ambient temperature of 26°C, and 28.7°C under ambient temperature of 32°C. It is important to note that when ambient temperature is 32°C, the high FF TEC is unable to cool the skin temperature to 30.5°C, which is typically the skin temperature in normal ambient condition of 22°C. This implies that in hot climate the high FF TEC is ineffective to cool the human body. The low FF and the ultra-low FF TECs, on the other hand, can be observed to cool the human skin below 29°C, when ambient temperature is 32°C, emphasizing the fact that these modules are quite effective even in extreme climate.

Fig. 8(a)-(c). Cold-side temperature for different TECs deployed on human body under different ambient temperatures. Increase in ambient temperature negatively impacts the cooling performance of the TECs. In a hot climate (ambient temperature $\sim 32^{\circ}\text{C}$), the high FF TEC is unable cool the skin temperature to its normal temperature ($\sim 30.5^{\circ}\text{C}$), whereas the low and ultra-low FF TECs are effective even in extreme climate. Circles illustrate the experimental data whereas solid lines show the modeling results.

(5) Instead of directly applying the designed TECs to human body test, experiments under a controllable environment (e.g., fixed temperature or heat flux conditions) should be completed in order to more accurately examine the TEC performance. The heat transfer rate of the fin and heat flux from skin should be measured in the experiments, or at least the estimates should be verified.

Author's reply: Thanks a lot for the suggestions. Based on the reviewer's recommendations, we have performed the experiments on different TEC modules under controlled environment and compared the outcomes with the results obtained for on-body tests. The controlled environment experiments were performed with a fixed temperature heat source and ambient as the heat sink.

A thermal resistor of known thermal conductivity (0.94 W/m-K) and dimensions: 16 mm × 18 mm × 10 mm was placed between the heat source and TEC to mimic the thermal resistance of the human skin. A transient study was performed where heat source temperature was fixed at 34°C and cold-side temperature of TECs was monitored over time. Figure 5(f)-(g) in the revised manuscript depict the experimental results obtained for different TECs under controlled environment and on human body. It can be noted that cold-side temperature and cooling (ΔT) obtained under controlled environment and on human body for different TECs follow the similar trend. At a fixed applied current, cold-side temperature of a TEC first decreases with time, reaches a minimum value, then slowly increases and finally saturates after about 10 mins. The transient minima are lower than steady-state cold-side temperature for all TECs. However, it is interesting to note that the ultra-low fill factor TEC generates least dynamic and static cold-side temperature and thus maximum cooling followed by the low fill factor TEC and the high fill factor TEC.

Following information has been added in the revised manuscript.

In order to mitigate the effect of human factors on experimental results, the experiments on different TEC modules were also performed under controlled environment and the outcomes were compared with the results obtained on human body. The controlled environment consisted of a heat source at fixed temperature (34°C) and a thermal resistor of known thermal conductivity (0.94 W/m-K) and dimensions: 16 mm × 18 mm × 10 mm placed between heat source and TEC to mimic the thermal resistance of the human skin. Fig. 5(f) and (g) depict the transient experimental data obtained for different FF TECs under controlled environment and on human body. It can be noted that cold-side temperature and cooling (ΔT) obtained under controlled environment and on human body for different TECs follow the similar trend.

Preliminary experiments revealed that the optimal electric current is ~ 1.0 A for the high FF TEC, ~ 2.4 A for the low FF TEC, and ~ 2.0 A for the ultra-low FF TEC. It can be noted that at a fixed applied current, cold-side temperature of a TEC first decreases with time, reaches a minimum value, then slowly increases and finally saturates after ~ 10 mins. The transient minima are lower than the steady-state cold-side temperature for all TECs. It is interesting to note that the ultra-low FF TEC generates the least cold-side temperature and thus maximum cooling followed by the low FF TEC and the high FF TEC.

Fig. 5(f)-(g). Transient study performed on different FF TECs under controlled environment and on human body. The transient minima are lower than steady-state cold-side temperature for all TECs. The ultra-low FF TEC generates least cold-side temperature and thus maximum cooling followed by low FF TEC and high FF TEC.

Figure 8(d)-(f) illustrate the cooling flux for different TECs deployed on human body under different ambient temperature. It can be noted that when the applied electric current is zero, depending upon the ambient temperature, the heat flux from human skin varies the range of 15-50 mW/cm^2 . The cooling flux increases with increase in electric current, but it decreases with

increases in ambient temperature. The peak cooling flux can be observed in the range of 85-110 mW/cm² for the high FF TEC, 120-140 mW/cm² for the low FF TEC, and 115-130 mW/cm² for the ultra-low FF TEC.

Following information has been added in the revised manuscript.

Fig. 8(d)-(f) illustrate the cooling flux for different TECs deployed on human body under different ambient temperatures. It can be noted that when the applied electric current is zero, depending upon the ambient temperature, the heat flux from human skin varies in the range of 15-50 mW/cm². The cooling flux increases with increase in electric current, but it decreases with increase in ambient temperature. The peak cooling flux can be observed in the range of 85-110 mW/cm² for the high FF TEC, 120-140 mW/cm² for the low FF TEC, and 115-130 mW/cm² for the ultra-low FF TEC. It has been suggested that if a localized thermal management system is able to remove 23 W of heat from human body, the cooling setpoint of household heating, ventilation, and air conditioning (HVAC) system can be increased by 2°C, leading to considerable saving in energy consumption⁵⁹. Considering the surface area of 1.8 m² for an average adult⁵⁹, it can be calculated that 23 W of body heat can be removed by covering 1.0-2.0% of the body surface with TECs.

Fig. 8(d)-(f). The numerical results illustrating cooling flux versus electric current for different FF TECs deployed on human body under different ambient temperatures. The cooling flux increases with increase in electric current, but it decreases with increases in ambient temperature. The peak cooling flux lies in the range of (a) 85-110 mW/cm² for High FF TEC, (b) 120-140 mW/cm² for low FF TEC, and (c) 115-130 mW/cm² for ultra-low FF TEC.

Reviewer #3 (Remarks to the Author):

(1) This paper focuses on the optimum design of Thermoelectric Coolers (TEC) operating under huge thermally resistive environment for localized wearable cooling applications. Nevertheless, the theoretical modeling and experiments conducted in this study were under the atmosphere temperature of 22 °C, which is not a typical hot condition requiring personal cooling. To demonstrate the cooling effect of TEC and potential energy saving in building HVAC systems through expansion of set-points as articulated in the introduction of this research, modeling and experiments should be carried out at an ambient temperature of above 26 °C (4 °F more than the normal set-points to achieve 20% saving in HVAC energy for cooling). Also, it is better to study the cooling effect of TEC at different ambient temperatures and show the influence of environment on its performance.

Authors' reply: Thanks much for the suggestions. Based on the reviewer's recommendations, we have performed numerical and experimental studies on different fill factor TECs deployed under different ambient temperature conditions. Figure 8 (a)-(c) in the revised manuscript show the cold-side temperature of different TECs. It can be noted that the initial skin temperature is higher at higher ambient temperature, indicating the fact that the human body temperature

increases with increase in ambient temperature. The average initial skin temperature was noted to be 30.6°C under ambient temperature of 22°C, 32.1°C under ambient temperature of 26°C, and 34.6°C under ambient temperature of 32°C. The increase in ambient temperature also has negative impact on the cooling effect of the TECs. When ambient temperature is 22°C, the high FF TEC generates minimum cold-side temperature of 25.2°C, which increases to 27.7°C under ambient temperature of 26°C, and 31.7°C under ambient temperature of 32°C. The low FF TEC generates minimum cold-side temperature of 22.4°C under ambient temperature of 22°C, 25°C under ambient temperature of 26°C, and 29°C under ambient temperature of 32°C. On the other hand, the ultra-low FF TEC generates minimum cold-side temperature of 22.3°C under ambient temperature of 22°C, 24.9°C under ambient temperature of 26°C, and 28.7°C under ambient temperature of 32°C. Please note that when ambient temperature is 32°C, the high fill factor TEC is unable to cool the skin temperature to 30.5°C, which is typically the skin temperature in normal ambient condition of 22°C. This infers that in hot climate the high FF TEC is ineffective in cooling the human body. The low and ultra-low FF TECs, on the other hand, can be observed to cool the human skin below 29°C, when ambient temperature is 32°C, emphasizing the fact that these modules are quite effective even in extreme climate.

Following information has been added in the revised manuscript.

It is important to note that higher cooling by the low and ultra-low FF TECs does not naturally result in a better thermal comfort. In fact, a large temperature drop on human skin can be quite uncomfortable; therefore, for practical purposes, users need to be provided with a temperature controller to control the cooling based on thermal preferences. None-the-less, since the low and ultra-low FF TECs proposed in this study have capability to generate a larger temperature drop

than the high FF TEC, they are expected to be more effective in hot-weather conditions. Fig. 8 (a)-(c) depict the cold-side temperature of the different TECs deployed on human body under different ambient temperatures. It can be noted that the initial skin temperature is higher at higher ambient temperature, indicating the fact that the human body temperature increases with increase in ambient temperature. The average initial skin temperature was noted to be 30.6°C under ambient temperature of 22°C, 32.1°C under ambient temperature of 26°C, and 34.6°C under ambient temperature of 32°C. In Fig. 8(a), when ambient temperature is 22°C, the high FF TEC generates minimum cold-side temperature of 25.2°C, which increases to 27.7°C under ambient temperature of 26°C, and 31.7°C under ambient temperature of 32°C. In Fig. 8(b), the low FF TEC generates minimum cold-side temperature of 22.4°C under ambient temperature of 22°C, 25°C under ambient temperature of 26°C, and 29°C under ambient temperature of 32°C. On the other hand, in Fig. 8(c), the ultra-low FF TEC module generates minimum cold-side temperature of 22.3°C under ambient temperature of 22°C, 24.9°C under ambient temperature of 26°C, and 28.7°C under ambient temperature of 32°C. It is important to note that when ambient temperature is 32°C, the high FF TEC is unable to cool the skin temperature to 30.5°C, which is typically the skin temperature in normal ambient condition of 22°C. This implies that in hot climate the high FF TEC is ineffective to cool the human body. The low FF and the ultra-low FF TECs, on the other hand, can be observed to cool the human skin below 29°C, when ambient temperature is 32°C, emphasizing the fact that these modules are quite effective even in extreme climate.

Fig. 8(a)-(c). Cold-side temperature for different TECs deployed on human body under different ambient temperatures. Increase in ambient temperature negatively impacts the cooling performance of the TECs. In a hot climate (ambient temperature $\sim 32^{\circ}\text{C}$), the high FF TEC is unable to cool the skin temperature to its normal temperature ($\sim 30.5^{\circ}\text{C}$), whereas the low and ultra-low FF TECs are effective even in extreme climate. Circles illustrate the experimental data whereas solid lines show the modeling results.

(2) Besides, to maintain thermal comfort in hot environment, TEC should remove a certain amount of heat from human body. Based on the FOA of ARPAC's DELTA program, an additional 23W of heat removal is required if the up-bound of neutral set-points is increased by 4 $^{\circ}\text{F}$. It is suggested to show heat removal the proposed TEC, given its covering area and weight.

Authors' reply: Thanks again for the suggestions. Figure 8(d)-(f) in the revised manuscript depict the heat flux for different fill factor TECs deployed on human body under different ambient temperatures. When the applied electric current is zero, depending upon the ambient temperature, the heat flux from human skin varies in the range of 15-50 mW/cm^2 . The cooling flux increases with increase in electric current, but it decreases with increase in ambient

temperature. The peak cooling flux can be observed to be in the range of 85-110 mW/cm² for the high FF TEC, 120-140 mW/cm² for the low FF TEC, and 115-130 mW/cm² for the ultra-low FF TEC. Please note that an average adult has a surface area of 1.8 m². That indicates that these TECs can remove a total of 1500-2500 W of body heat under ambient temperature of 22-32°C. Of course, this is not practical as it would require covering the entire human body with TECs. However, in order to achieve the target value of 23 W, 1.0-2.0% of the body surface would be enough.

Following information has been added in the revised manuscript.

Fig. 8(d)-(f) illustrate the cooling flux for different TECs deployed on human body under different ambient temperatures. It can be noted that when the applied electric current is zero, depending upon the ambient temperature, the heat flux from human skin varies in the range of 15-50 mW/cm². The cooling flux increases with increase in electric current, but it decreases with increase in ambient temperature. The peak cooling flux can be observed in the range of 85-110 mW/cm² for the high FF TEC, 120-140 mW/cm² for the low FF TEC, and 115-130 mW/cm² for the ultra-low FF TEC. It has been suggested that if a localized thermal management system is able to remove 23 W of heat from human body, the cooling setpoint of household heating, ventilation, and air conditioning (HVAC) system can be increased by 2°C, leading to considerable saving in energy consumption⁵⁹. Considering the surface area of 1.8 m² for an average adult⁵⁹, it can be calculated that 23 W of body heat can be removed by covering 1.0-2.0% of the body surface with TECs.

Fig. 8(d)-(f). The numerical results illustrating cooling flux versus electric current for different FF TECs deployed on human body under different ambient temperatures. The cooling flux increases with increase in electric current, but it decreases with increase in ambient temperature. The peak cooling flux lies in the range of (a) 85-110 mW/cm² for high FF TEC, (b) 120-140 mW/cm² for low FF TEC, and (c) 115-130 mW/cm² for ultra-low FF TEC.

(3) Furthermore, the large temperature reduction in localized skin area covered by the TEC can cause severe localized cold discomfort. It is advised to evaluate the local cold discomfort.

Authors' reply: We agree that the high cooling does not naturally produce better thermal comfort. In fact, a very large temperature drop on human skin could be quite uncomfortable. However, as stated in FOA of ARPA-E DELTA program, one of the key advantages of using a localized cooling thermal management system is that it enables users to adjust the cooling based on their thermal comfort. For practical purposes, therefore, the users need to be provided with a temperature controller to control the temperature drop as per the desired comfort level.

We have added following explanation in the revised manuscript.

It is important to note that higher cooling by the low and ultra-low FF TECs does not naturally result in a better thermal comfort. In fact, a large temperature drop on human skin can be quite uncomfortable; therefore, for practical purposes, users need to be provided with a temperature controller to control the cooling based on thermal preferences. None-the-less, since the low and ultra-low FF TECs proposed in this study have capability to generate a larger temperature drop than the high FF TEC, they are expected to be more effective in hot-weather conditions. Fig. 8 (a)-(c) depict the cold-side temperature of the different TECs deployed on human body under different ambient temperatures. It can be noted that the initial skin temperature is higher at higher ambient temperature, indicating the fact that the human body temperature increases with increase in ambient temperature. The average initial skin temperature was noted to be 30.6°C under ambient temperature of 22°C, 32.1°C under ambient temperature of 26°C, and 34.6°C under ambient temperature of 32°C. In Fig. 8(a), when ambient temperature is 22°C, the high FF TEC generates minimum cold-side temperature of 25.2°C, which increases to 27.7°C under ambient temperature of 26°C, and 31.7°C under ambient temperature of 32°C. In Fig. 8(b), the low FF TEC generates minimum cold-side temperature of 22.4°C under ambient temperature of 22°C, 25°C under ambient temperature of 26°C, and 29°C under ambient temperature of 32°C. On the other hand, in Fig. 8(c), the ultra-low FF TEC module generates minimum cold-side temperature of 22.3°C under ambient temperature of 22°C, 24.9°C under ambient temperature of 26°C, and 28.7°C under ambient temperature of 32°C. It is important to note that when ambient temperature is 32°C, the high FF TEC is unable to cool the skin temperature to 30.5°C, which is typically the skin temperature in normal ambient condition of 22°C. This implies that in hot climate the high FF TEC is ineffective to cool the human body. The low FF and the ultra-low FF TECs, on the other hand, can be observed to cool the human skin below 29°C, when ambient

temperature is 32°C, emphasizing the fact that these modules are quite effective even in extreme climate.

Fig. 8(a)-(c). Cold-side temperature for different TECs deployed on human body under different ambient temperatures. Increase in ambient temperature negatively impacts the cooling performance of the TECs. In a hot climate (ambient temperature ~32°C), the high FF TEC is unable cool the skin temperature to its normal temperature (~30.5°C), whereas the low and ultra-low FF TECs are effective even in extreme climate. Circles illustrate the experimental data whereas solid lines show the modeling results.

(4) Moreover, without an effective heat rejection in a hotter environment than the ambient temperature (22 °C) investigated in this study, it is advised to show whether the cooling effect is stable.

Author's reply: Thanks a lot for the suggestions. Based on the reviewer's recommendations, a transient study was performed where a fixed electric current was applied and cold-side temperature of TECs was monitored over time. Figure 5(f) and (g) in the revised manuscript depict the experimental results indicating cold-side temperature and cooling obtained for

different TECs over time. Based on reviewer # 2 recommendations, experimental results under a controlled environment have been also added in Figure 5(f) and (g) for comparison. It can be noted that under a fixed applied current, cold-side temperature of a TEC first decreases with time, reaches a minimum value, then slowly increases and finally saturates after about 10 mins. The transient minima are lower than steady-state cold-side temperature for all TECs. However, it is interesting to note that the ultra-low FF TEC generates least cold-side temperature and thus maximum cooling followed by the low FF TEC and the high FF TEC. The steady state cold-side temperature can be noted to be 26.3°C for the high FF TEC, 24.3°C for the low FF TEC and 22.9°C for the ultra-low FF TEC.

Following information has been added in the revised manuscript.

In order to mitigate the effect of human factors on experimental results, the experiments on different TEC modules were also performed under controlled environment and the outcomes were compared with the results obtained on human body. The controlled environment consisted of a heat source at fixed temperature (34°C) and a thermal resistor of known thermal conductivity (0.94 W/m-K) and dimensions: 16 mm × 18 mm × 10 mm placed between heat source and TEC to mimic the thermal resistance of the human skin. Fig. 5(f) and (g) depict the transient experimental data obtained for different FF TECs under controlled environment and on human body. It can be noted that cold-side temperature and cooling (ΔT) obtained under controlled environment and on human body for different TECs follow the similar trend. Preliminary experiments revealed that the optimal electric current is ~1.0 A for the high FF TEC, ~2.4 A for the low FF TEC, and ~2.0 A for the ultra-low FF TEC. It can be noted that at a fixed applied current, cold-side temperature of a TEC first decreases with time, reaches a minimum value, then slowly increases and finally saturates after ~10 mins. The transient minima are lower

than the steady-state cold-side temperature for all TECs. It is interesting to note that the ultra-low FF TEC generates the least cold-side temperature and thus maximum cooling followed by the low FF TEC and the high FF TEC.

Fig. 5(f)-(g). Transient study performed on different FF TECs under controlled environment and on human body. The transient minima are lower than steady-state cold-side temperature for all TECs. The ultra-low FF TEC generates least cold-side temperature and thus maximum cooling followed by low FF TEC and high FF TEC.

REVIEWERS' COMMENTS:

Reviewer #1 (Remarks to the Author):

The authors have made necessary amendments and I am satisfied with the current presence of the paper, and thus, suggest to accept it for publication.

Reviewer #2 (Remarks to the Author):

The authors have addressed most of the reviewer comments given in last round of review. Few minor revisions are necessary before publication:

(1) Manuscript page 3, "The wearable TECs, therefore, experience a very low heat load and an extremely high thermally resistive environment." Some quantitative numbers or even a summary table for comparison of the extremely high thermally resistive environment will be helpful.

(2) Manuscript page 4, "The thermoelectric modules with fill factor less than 20% have been recommended for on-body applications, which brings challenges in terms of mechanical stability." Will the TE modules fabricated in this work face the similar problem because of low and ultra-low FF factors?

(3) Manuscript page 4, "The ultra-low fill fraction module proposed in this paper exhibits 170% higher cooling,..." Higher cooling power or temperature drop? Than commercial TEC?

(4) Manuscript page 11, reference(s) is needed to support the claim "For body cooling, however, cold-side temperature drop (ΔT) can be considered a more appropriate criterion to judge the effectiveness of TECs as temperature change can be easily perceived by the human body."

Reviewer #3 (Remarks to the Author):

This paper provided a comprehensive study on the optimum design of Thermoelectric Coolers for wearable cooling applications. The author has tried to address my concerns about cooling power and local thermal comfort of such a device, which are crucial to the potential of such a concept. Based on their experimental work and theoretical modelling under the condition that there is close contact between the device's cold side and human skin, the maximum cooling power of different designs ranges from 85~140 mW/cm². To achieve 23W cooling required for expanding the indoor temperature by 2 °C, the required cooling area would be between 160-270 cm². Although this is just about 1~2% of the body surface area, but it is substantial, considering one can only use the TC to cover the possibly exposed lower arm and leg areas where one can withstand relatively lower temperature. Furthermore the weight of the TC would also be a major concern in practical applications. a 2 in² (12 cm²) TC assembly may weigh 100g. 160~270 cm² would weigh 1.3-2.3 kg. High weight, large impermeable and inflexible area covering the body surface and local cold discomfort may prohibit its practical application. If one uses just one of such TC at the wrist, it only provide 1~2W cooling. Furthermore, I wish to know the heat sink temperature at the peak cooling power under different environmental temperature. For safety reasons, the maximum heat sink temperature should be less than 42 °C. This can be an issue when it is used in hot environment.

Reviewers' comments:

Reviewer #1 (Remarks to the Author):

The authors have made necessary amendments and I am satisfied with the current presence of the paper, and thus, suggest to accept it for publication.

Author's reply: Thanks, and we much appreciated your comments and suggestions.

Reviewer #2 (Remarks to the Author):

The authors have addressed most of the reviewer comments given in last round of review. Few minor revisions are necessary before publication:

(1) Manuscript page 3, "The wearable TECs, therefore, experience a very low heat load and an extremely high thermally resistive environment." Some quantitative numbers or even a summary table for comparison of the extremely high thermally resistive environment will be helpful.

Author's reply: Thanks a lot for the suggestion. A comparison table listing the heat transfer coefficient under different operating conditions is provided in the Supplementary information.

We have added the heat transfer coefficient for human skin reported in the literature.

Supplementary Table 2. Typical values of heat transfer coefficient (h)^{16,17}

Type of convection	h (W/m ² -K)
Free convection of gas	2-25
Free convection of liquid	10-1000
Forced convection of gases	25-250

Forced convection of liquids	50-20,000
Boiling and condensation	2500-100,000
Human skin	20-100

(2) Manuscript page 4, " The thermoelectric modules with fill factor less than 20% have been recommended for on-body applications, which brings challenges in terms of mechanical stability." Will the TE modules fabricated in this work face the similar problem because of low and ultra-low FF factors?

Author's reply: The fabrication of low fill factor thermoelectric module certainly requires changes in manufacturing steps. However, with advancements in automated drop-n-reflow technology, we think that these changes can be accommodated. In this study, the optimum TEC modules were fabricated using dies that guide the placement of legs and provide support structure. This results in better alignment and mechanical properties of modules.

(3) Manuscript page 4, "The ultra-low fill fraction module proposed in this paper exhibits 170% higher cooling,..." Higher cooling power or temperature drop? Than commercial TEC?

Author's reply: The ultra-low fill fraction module proposed in this paper exhibits 170% higher temperature cooling than the commercial TEC. We have modified the sentence in the paper.

(4) Manuscript page 11, reference(s) is needed to support the claim "For body cooling, however, cold-side temperature drop (ΔT) can be considered a more appropriate criterion to judge the effectiveness of TECs as temperature change can be easily perceived by the human body."

Author's reply: Thanks for the suggestion. We have added following information in the paper along with the references.

Prior studies have reported that the human skin on the certain part of human hand can perceive the temperature differential as low as 0.20°C for warming (at a rate of $2.1^{\circ}\text{C}\text{s}^{-1}$) and 0.11°C for cooling (at a rate of $1.9^{\circ}\text{C}\text{s}^{-1}$)^{53,54}.

References:

[53] Jones, Lynette A., and Hsin-Ni Ho. "Warm or cool, large or small? The challenge of thermal displays." *IEEE Transactions on Haptics* 1.1 (2008): 53-70.

[54] Joseph, Stevens C., and Choo, Kenneth K. "Temperature sensitivity of the body surface over the life span." *Somatosensory & motor research* 15.1 (1998): 13-28.

Reviewer #3 (Remarks to the Author):

This paper provided an comprehensive study on the optimum design of Thermoelectric Coolers for wearable cooling applications. The author has tried to address my concerns about cooling power and local thermal comfort of such a device, which are crucial to the potential of such a concept.

(1) Based on their experimental work and theoretical modelling under the condition that there is close contact between the device's cold side and human skin, the maximum cooling power of

different designs ranges from 85~140 mW/cm². To achieve 23 W cooling required for expanding the indoor temperature by 2°C, the required cooling area would be between 160-270 cm². Although this is just about 1~2% of the body surface area, but it is substantial, considering one can only use the TC to cover the possibly exposed lower arm and leg areas where one can withstand relatively lower temperature.

Author's reply: Table R1 below illustrates the calculations for the TEC area needed for the removal of 23 W of body heat. It can be noted that low and ultra-low FF TECs require ~180 cm² of the skin coverage, which is ~1% of the human body. The mean surface area of one hand of an average human adult is ~540 cm² [1]. This indicates that the total area needed by TECs is not substantially large.

We have added this information in the revised Supplementary information.

Table R1. Calculations illustrating the TEC area needed for the removal of 23 W of body heat.

	TEC base area (cm²)	Avg. cooling (mW/cm²)	TEC area needed for 23 W (cm²)	% area of human body
High FF TEC	2.56	97.5	235.9	1.31%
Low FF TEC	2.56	130.0	176.9	0.98%
Ultra-low FF TEC	2.56	122.5	187.7	1.04%

(2) Furthermore the weight of the TC would also be a major concern in practical applications. a 2 in2 (12 cm²) TC assembly may weigh 100g. 160~270 cm² would weigh 1.3-2.3 kg. High weight, large impermeable and inflexible area covering the body surface and local cold discomfort may

prohibit its practical application. If one uses just one of such TC at the wrist, it only provide 1~2W cooling.

Author’s reply: Table R2 below illustrates the calculations for the TEC weight (including heat sink) needed for the removal of 23 W of body heat. It can be noted that low and ultra-low FF TECs weigh ~500 gm, which is within the range of the weight of clothes and other day-to-day wearables shown in Table R3 [2-3].

We have added this information in the revised Supplementary information.

Table R2. Calculations illustrating the TEC weight needed for the removal of 23 W of body heat.

	Weight of a TEC with heat sink (gm)	TEC base area (cm²)	Avg. cooling (mW/cm²)	Weight of TECs for 23 W of cooling (gm)
High FF TEC	7.9	2.56	97.5	728
Low FF TEC	7.1	2.56	130.0	491
Ultra-low FF TEC	6.9	2.56	122.5	506

Table R3. Average weight of men’s clothes and other day-to-day wearables [2-3].

Wearables	Weight (gm)
Sports shirt, T-shirt	220 - 300
Shorts	250 - 350
Jersey	450 - 600
Pants	600 - 700
Jeans	650 - 800
Business suit	1200 - 1800
Sports suit	1000 - 1300

Slippers	~400
Sandals	~450
Running shoes, trainers	~400
Casual sneakers	~500
Shoes, Gumboots	~600
Handwatch	~300

(3) Furthermore, I wish to know the heat sink temperature at the peak cooling power under different environmental temperature. For safety reasons, the maximum heat sink temperature should be less than 42°C. This can be an issue when it is used in hot environment.

Author's reply: The heat sink temperature at the optimal current for maximum cooling for our TECs were noted to be 45-55°C, which is bit higher than the threshold value suggested by the reviewer. However, it should be noted that the hot-side temperature of the TEC can be substantially reduced by modifying the heat sink, as shown in Figure R1. In this study, we have used an off-the-shelf heat sink purchased from McMaster-Carr [4]. As shown on Figure R1 (a), the maximum hot-side temperature in this case is ~ 47°C. Temperature contours illustrated in Figure R1(b)-(d) shows that the hot-side temperature decreases by increasing the number of fins and adjusting the fin dimensions. It is also interesting to note that lower hot-side temperature and thus lower temperature difference between hot- and cold-sides of TEC results in higher cooling due to reduced reverse thermal current.

Figure R1. Temperature distribution in the heat sink and TEC. Hot-side temperature of the TEC can be substantially reduced by optimizing the heat sink.

References

- [1] Kaye, R., & Konz, S. (1986). Volume and Surface Area of the Hand. *Proceedings of the Human Factors Society Annual Meeting*, 30(4), 382–384.
- [2] <https://rocketmf.com/en/weight#tab-1>
- [3] https://www.parcl.com/education/customers/shipping_weight/
- [4] <https://www.mcmaster.com/8822t11>